

# Bootstrapping frustrated magnets:
# The fate of the chiral O(N) × O(2) universality class

**Marten Reehorst[1,2⋆], Slava Rychkov[3†], Benoit Sirois[3,4‡] and Balt C. van Rees[1∘]**

**1** CPHT, CNRS, Ecole Polytechnique, Institut Polytechnique de Paris, Palaiseau, France
**2** Department of Mathematics, King's College London, Strand, London, WC2R 2LS, UK
**3** Institut des Hautes Études Scientifiques, 91440 Bures-sur-Yvette, France
**4** Laboratoire de Physique de l'École normale supérieure, ENS, Université PSL, CNRS,
Sorbonne Université, Université de Paris, F-75005 Paris, France

⋆ marten.reehorst@kcl.ac.uk , † slava@ihes.fr , ‡ benoit.sirois@umontreal.ca ,
∘ balt.van-rees@polytechnique.edu

## Abstract

We study multiscalar theories with O(N) × O(2) symmetry. These models have a stable fixed point in $d$ dimensions if $N$ is greater than some critical value $N_c(d)$. Previous estimates of this critical value from perturbative and non-perturbative renormalization group methods have produced mutually incompatible results. We use numerical conformal bootstrap methods to constrain $N_c(d)$ for $3 \leqslant d < 4$. Our results show that $N_c > 3.78$ for $d = 3$. This favors the scenario that the physically relevant models with $N = 2, 3$ in $d = 3$ do not have a stable fixed point, indicating a first-order transition. Our result exemplifies how conformal windows can be rigorously constrained with modern numerical bootstrap algorithms.

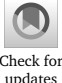

# 1   Introduction

It has been known for decades now [1] that a multiscalar theory with $O(N) \times O(2)$ global symmetry describes the critical modes of $N$-component classical stacked triangular antiferromagnets (STAs) and helimagnets (see App. A for a brief review). The Hamiltonian is [1]:

$$\mathcal{H} = \partial_\mu \varphi \cdot \partial_\mu \varphi^* + m^2 \varphi \cdot \varphi^* + u(\varphi \cdot \varphi^*)^2 + v(\varphi \cdot \varphi)(\varphi^* \cdot \varphi^*), \tag{1}$$

where $\varphi$ is an $N$-component complex field.[1] The basic question is:

> *For the physical values $N = 2, 3$ and in three dimensions, does the model* (1)
> *have a stable fixed point of the Renormalization Group (RG) flow?*
>       (2)

If yes, for materials within the basin of attraction of this fixed point, experiments should observe second-order phase transitions. In the opposite case, all phase transitions should be first order.

---

[1]In this representation, $O(N)$ acts as $\varphi \to R\varphi$, while $O(2)$ acts as $\varphi \to e^{i\alpha}\varphi$ and $\varphi \leftrightarrow \varphi^*$.

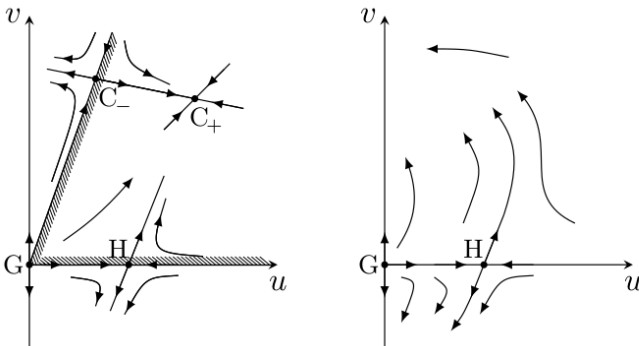

Figure 1: RG flow and fixed point of the model (1) in $d = 4 - \epsilon$ dimensions, for $N > N_c(d)$ (left) and $N < N_c(d)$ (right). Figure from [1].

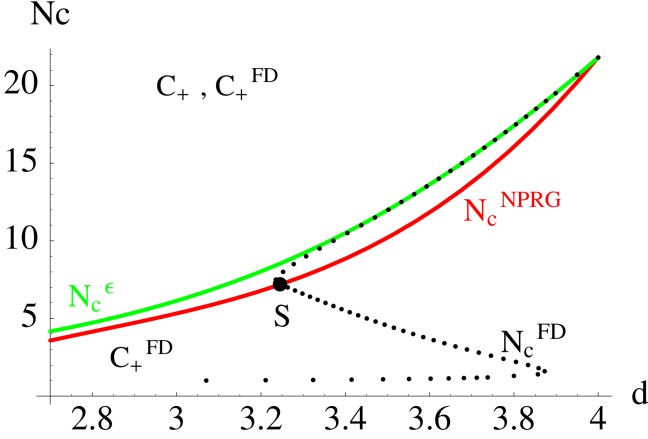

Figure 2: Comparison between NPRG, fixed-dimensional RG and the $\epsilon$-expansion. Figure from [10].

Although the model and the question are simple enough, the subject has been the arena of a controversy [2–4], with various methods giving contradictory results, as we now review.

A time-honored way to study the critical properties of multiscalar models is through the $\epsilon$-expansion around $d = 4$ [5]. Applied to our model, the $\epsilon$-expansion predicts the existence of a critical curve $N_c(d)$ above which a stable fixed point exists. This stable fixed point is called chiral and we will denote it as $\mathcal{C}_+$ in Fig. 1. At $N = N_c(d)$, the chiral fixed point merges and annihilates with an unstable fixed point $\mathcal{C}_-$, called antichiral.[2] Below this collision, only two fixed points remain, both unstable: the Gaussian $\mathcal{G}$ and the $O(2N)$ invariant "Heisenberg" fixed point $\mathcal{H}$, which has $v = 0$ in (1). These RG flows at $N > N_c$ and $N < N_c$ are illustrated in Fig. 1. Perturbation theory allows to compute $N_c(d = 4 - \epsilon)$ as an asymptotic series expansion in $\epsilon$, known up to 6 loops [6].[3] Using various resummation techniques, [6] concludes that $N_c(3) = 5.96(19)$, well above the physical values. To summarize, the $\epsilon$-expansion predicts first-order phase transitions for $N = 2, 3$.[4]

---

[2]The "chiral fixed point" refers to chirality present in low-T ground states of STAs, see App. A. The "antichiral fixed point" refers not to opposite chirality of the ordered phase (both chiralities may be present), but to the possibility of annihilation with the chiral fixed point.

[3]The very first RG studies of the bifundamental scalar model with $O(M) \times O(N)$ symmetry, at the lowest order in the $\epsilon$-expansion, were performed in mid-1970's in [7–9].

[4]The $\epsilon$-expansion also predicts further changes to the phase diagram below $N = 2$, see [1] and footnote 22 for more details. We will not be concerned with these transitions.

A competing theoretical view is that of the perturbative RG performed directly in $d = 3$ [11–14] (fixed-dimensional (FD) RG). For this method, $\beta$-functions are expanded in powers of the quartic couplings of the model in the dimension of interest. The $\beta$-functions are then resummed before looking for a fixed point. This may be contrasted to the $\epsilon$-expansion, where fixed points and critical exponents are found in the perturbative regime as series-expansions in $\epsilon$, and resummation is performed at the last step when extrapolating the exponents to the desired dimension. For the simpler $O(N)$ model these two methods agree, however they do not agree very well for $O(N) \times O(2)$. The fixed-dimensional RG predicted the region of the $(d, N)$ plane where a stable chiral fixed point exists to be bounded by an S-shaped curve which is not a single-valued function $N_c(d)$, see Fig. 2. Although the boundary agrees near $d = 4$ with the $\epsilon$-expansion, it deviates and takes a couple of turns as $N$ is lowered. The resulting region where a stable chiral fixed point exists is significantly larger than for the $\epsilon$-expansion. Furthermore, the new stable fixed points found within FD schemes but not in the $\epsilon$-expansion turn out to be "of focus type", meaning that they have complex correction-to-scaling exponents. This is in conflict with the unitarity of the model, requires unlikely crossings in the operator spectrum, and contradicts basic principles of renormalization group such as the gradient flow property.[5] See Appendix B for a more detailed discussion.

Unfortunately, experimental data and Monte Carlo simulations do little to clear up the picture. Some Monte Carlo results show clear signs of first-order transitions [20, 21] while some claim to confirm the existence of focus-type fixed points [22].[6] Experiments are plentiful, but the consensus is that there is such a large discrepancy of critical exponents in the cases where a second-order phase transition is believed to happen, that it is impossible to discern between true second-order and weak first-order behavior [2, 3]. We are left with an unfortunate situation where the most standard methods fail to give a coherent picture, and thus new perspectives from alternative methods are necessary.

In this respect, calculations have also been performed using Functional, or Nonperturbative RG (NPRG). The results rather confirm the picture of the $\epsilon$-expansion [3, 23], producing a curve $N_c(d)$ in a reasonably good agreement with the $\epsilon$-expansion all the way down to three dimensions, see Fig. 2.

In this paper we will approach the problem using another nonperturbative method - the numerical conformal bootstrap.[7] Since its inception [26], thanks to many technical and conceptual improvements such as [27–38], the method has achieved several important results, notably the high precision determination of the critical exponents of the 3D Ising, O(2) and O(3) universality classes [39, 40]. It has already contributed to the resolution of some puzzles such as the liquid helium heat capacity anomaly [41] and the cubic instability of the Heisenberg magnets [42]. See [43–46] for reviews and results for other models, including with gauge fields and fermions.

Conformal bootstrap relies on the basic fact that a fixed point of the renormalization group flow is normally described by a unitary conformal field theory (see Appendix C), as well as on some assumptions about gaps in the operator spectrum. This represents a very different set of assumptions as compared to the renormalization group studies. In particular, the conformal bootstrap analysis does not suffer from truncation and resummation ambiguities inherent

---

[5]The RG flow of multiscalar models is widely expected to be a gradient flow. This has been long known to be true to three loops [15,16], and recently has been verified at five loops (and even six loops with some assumptions) [17], using the six-loop beta function results from [18,19]. Refs. [11–14] resummed each component of the beta-function separately - a procedure which may not have preserved the gradient property.

[6]It should be noted that the lattice model in [22] is reflection positive, rigorously excluding complex critical exponents (see Appendix B.1). This is in stark conflict with the complex correction-to-scaling exponent $\omega = 0.1^{+0.4}_{-0.05} + i\, 0.7^{+0.1}_{-0.4}$ they report.

[7]Previously, conformal bootstrap has already been used to study this problem in Refs. [24,25]. There are some methodological differences between our work and the approaches of these prior works, whose review is postponed to Section 4.

to perturbative and nonperturbative RG techniques mentioned above. It is therefore particularly interesting to inquire what the conformal bootstrap has to say about the $O(N) \times O(2)$ controversy.

For the purposes of conformal bootstrap analysis, question (2) is reformulated as:

> *For $N = 2, 3$ and in $d = 3$, is there a unitary conformal field theory (CFT) with $O(N) \times O(2)$ symmetry and with one relevant singlet scalar operator in its spectrum?* (3)

The latter condition is how the fixed point stability is translated into the CFT language, the only relevant singlet being the mass term in (1). Further conditions on the CFT to allow its identification with the RG fixed point of (1), such as a low-lying bifundamental scalar, will be discussed in Section 2.

Answering (3) directly is out of reach with the current state of the art. Instead we will address the question:

> *What is the shape of the critical curve defining $N_c(d)$ as we lower the spacetime dimension $d$ from 4 to 3?* (4)

Recall that we have perturbative control near $d = 4$ (as well as at large $N$ for any $d$), so the existence of the curve is uncontroversial. But could it be that the $\epsilon$-expansion and NPRG are wrong, and the boundary of the region where the stable fixed point exists is indeed S-shaped as in FD-studies [11–14]? We would like to be open-minded about this, even though another aspect of the FD-studies - focus fixed points - is ruled out as contradicting above-mentioned general principles.

Our strategy for investigating (4) is as follows. After making some minimal assumptions we will find a small allowed island in the space of CFT data that matches all the expectations for the $\mathcal{C}_+$ fixed point for large $N$. As we then lower $N$ this island varies in size, but eventually it shrinks and disappears entirely at some numerically determined value $N_c^{\mathrm{CB}}(d)$. So if one believes that the $\mathcal{C}_+$ fixed point was indeed located inside this island then the conclusion is unavoidably that $N_c^{\mathrm{CB}}(d)$ provides a rigorous lower bound for $N_c(d)$.[8]

In Fig. 3 we show $N_c^{\mathrm{CB}}(d)$ for $3 \leqslant d \leqslant 3.8$. The main salient features of this plot are as follows:

- The bootstrap curve $N_c^{\mathrm{CB}}(d)$ shows the same single-valued monotonic behavior as the $\epsilon$-expansion and NPRG curves. There is no sign of a turnaround similar to the one reported in FD-studies.

- At $d = 3$, we have $N_c^{\mathrm{CB}}(3) = 3.78$. This is quite a bit lower than the $\epsilon$-expansion and NPRG predictions, but above the physical values $N = 2, 3$.

The use of a vanishing island is the most natural way to determine the endpoint of a conformal window with the numerical conformal bootstrap. We expect it to be useful also in many other cases.

The paper is structured as follows. In Section 2 we describe our bootstrap setup, detailing in particular in Section 2.3 our choice of gap assumptions in important symmetry channels. We then move on to the detailed description of our bootstrap study and results in Section 3. In Section 4 we review previous bootstrap studies of the same problem [24, 25]. In Section 5 we conclude. In particular, we discuss how our results about the shape of the boundary curve relate to the question (3) about the existence of $N = 2, 3$ CFTs. We also have several appendices. Appendix A is dedicated to introducing frustrated noncollinear magnets, Appendix B critically

---

[8]With more computational resources we would obtain a smaller island and therefore potentially a disappearance for $N$ greater than $N_c^{\mathrm{CB}}(d)$. This is why $N_c^{\mathrm{CB}}(d)$ is a lower bound.

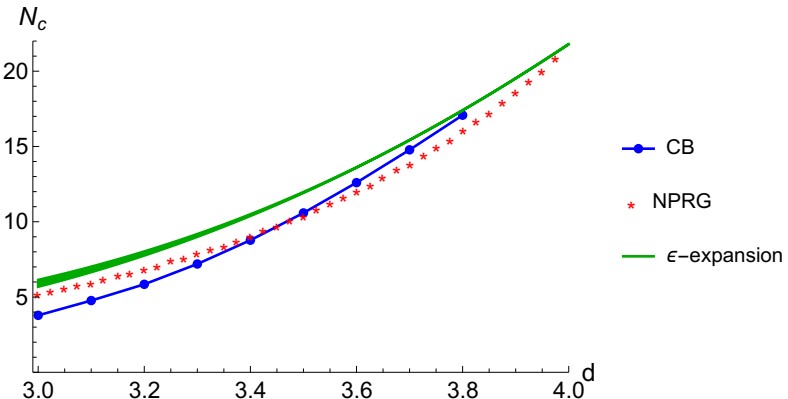

Figure 3: This plot, the main result of our paper, reports $N_c^{\text{CB}}(d)$ extracted by the conformal bootstrap (blue dots, joined by an interpolating blue curve). We also show the results of NPRG [3, Fig. 9] (red asterisks) and of the $\epsilon$-expansion (green band, whose width is the uncertainty obtained as in Appendix E.1). Conformal bootstrap results used spectrum gap assumptions (7), (9), (10) and the derivative order $\Lambda = 31$.

reviews the controversial notion of focus fixed points, and Appendix C reviews why scale invariance is expected to be enhanced to conformal invariance for our models. Appendix D is devoted to the calculation of the scaling dimension at $O(1/N)$ of a certain crucial operator, and Appendix E discusses results obtained in the $\epsilon$-expansion. Details of the numerical methods are collected in Appendix F.

In the ancillary mathematica notebook *EFM_Spectrum_Along_Our_Curves.nb*, we give the full spectrum and OPE coefficients extracted through the extremal functional method [31] along the $\Lambda = 31$ curves of Figs. 3, 8 and 9. The notebook explains the formatting of this data and provides example plots and tables to help visualize it.

We will be assuming that the reader is reasonably familiar with the conformal bootstrap philosophy and standard technology. We refer to the reviews [43–46] and to the pedagogical lecture notes [47, 48]. We will be also using more modern tools such as the navigator function [49] and the skydiving algorithm [50].

## 2 CFT setup

We are interested in the existence of the stable RG fixed point $\mathcal{C}_+$ of (1) with $O(N) \times O(2)$ global symmetry. The scale invariance of the fixed point is expected to be enhanced to conformal invariance, as we briefly review in Appendix C. We can therefore use the language of CFTs and the tools of the numerical conformal bootstrap. Instead of focusing only on the physically interesting cases of $d = 3$ and $N = 2, 3$, we will explore the wider range $3 \leqslant d \leqslant 4$ and $N \geqslant 2$. This includes the regimes $d \to 4$ and $N \to \infty$ which are controlled by the $\epsilon$-expansion and the $1/N$ expansion.

Primary CFT operators are characterized by dimension $\Delta$, spin $\ell$, and by a global symmetry irrep $\mathcal{R}$, which in our case are labeled XY, where X is an irrep of $O(N)$ and Y of $O(2)$. In the rest of this paper we will be using real notation in which the complex vector field $\varphi$ of (1) is repackaged as a real matrix field $\phi = \phi_{ai}$ ($a = 1, \ldots, N$, $i = 1, 2$), in the bifundamental (VV) representation of $O(N) \times O(2)$.[9]

---

[9]The complex field $\varphi$ in (1) is $\varphi_a = \phi_{a1} + i\phi_{a2}$.

Our study will use crossing symmetry constraints for the correlation functions

$$\langle \phi\phi\phi\phi \rangle, \qquad \langle \phi\phi ss \rangle, \qquad \langle ssss \rangle, \tag{5}$$

where $\phi$ and $s$ are the lowest-lying scalar primaries in the bifundamental (VV) and the total singlet (SS) irreps. In the Lagrangian description we have $s = \phi^2$, but we will not use this notation. The operator $s$ is relevant. All subsequent total singlet scalars have to be irrelevant – this is the RG fixed point stability condition.

Operators in the operator product expansion (OPE) $\phi \times s$ transform as VV. In the Lagrangian description, the VV scalars after $\phi$ are the cubic operators $\phi_{ai}\phi^2$ and $\phi_{aj}\phi_{bi}\phi_{bj}$. One linear combination of these, the derivative of the fixed point potential with respect to $\phi_{ai}$, is a descendant of $\phi$.[10] The orthogonal linear combination is a primary, denoted $\phi'$.[11] Made out of three scalar fields in the perturbative notation, this primary is expected to be relevant, while the subsequent primaries in the VV sector, made of 5 or more $\phi$'s, are expected to be irrelevant.

Operators in the $\phi \times \phi$ OPE have $X, Y = S, T, A$, where T, A are the symmetric traceless and antisymmetric 2-index tensor irreps. The important operators in this OPE include the stress tensor $T_{\mu\nu}$, which is a spin-2 SS primary of dimension $\Delta = d$, and the conserved currents of the $O(N)$ and $O(2)$ symmetries, which are spin-1 primaries transforming as AS and SA respectively, of dimension $\Delta = d - 1$. Below we will specify gap assumptions above these operators.

A recurrent theme of our study will be how to distinguish our CFT of interest $\mathcal{C}_+$ from the Heisenberg CFT $\mathcal{H}$, which has a larger symmetry $O(2N)$. This is a crucial issue since $\mathcal{H}$ is present below $N_c(d)$, and thus has the potential to pollute our bootstrap analysis. A particularly elegant way to disentangle the two theories in the correlator system Eq. (5) turns out to be the ST scalar channel in the $\phi \times \phi$ OPE, as we now proceed to discuss.

In model (1), the lowest operator in this channel is $\mathcal{O}_{ST} = \phi_{ai}\phi_{aj} - $ trace. At leading order in the $1/N$ expansion it has dimension close to 2 (in any $d$) for $\mathcal{C}_+$, whereas it has dimension $d - 2$ in $\mathcal{H}$. Since this difference is fairly big, a gap assumption in this channel might allow us to distinguish between both theories. In practice we have enough computational power to assume the existence of $\mathcal{O}_{ST}$ and impose a gap to the subleading primary instead. This operator also has different large-$N$ behavior in $\mathcal{H}$ and $\mathcal{C}_+$:

$$\Delta_{ST'} = \begin{cases} 4 + O(1/N), & \text{in } \mathcal{C}_+, \\ d + O(1/N), & \text{in } \mathcal{H}. \end{cases} \tag{6}$$

This is explained in more detail in Appendix D, where also the $O(1/N)$ terms are given. We see that, for $d < 4$ and for $N$ sufficiently large, $\Delta_{ST'}$ is expected to be larger in $\mathcal{C}_+$ than in $\mathcal{H}$. The simplest thing to do would be to put a carefully chosen gap above $\mathcal{O}_{ST}$, chosen precisely so that we exclude $\mathcal{H}$ but include $\mathcal{C}_+$. It turns out that we can do even better: with a smaller gap in the ST channel, which allows both $\mathcal{H}$ and $\mathcal{C}_+$, we are able to isolate both theories into separate islands. This will be enough for all practical purposes because we can track the disappearance of the $\mathcal{C}_+$ fixed point without interference from the $\mathcal{H}$ fixed point. The usefulness of the ST channel for this purpose is one of the main discoveries of our work.

To summarize, in Table 1 we list operators treated in our study as isolated. The $\phi$ and $s$ appear as both external and internal (i.e. exchanged) operators, the rest only as internal. As we explain in more detail below, we will numerically explore the three-dimensional parameter space corresponding to the varying scaling dimensions $\Delta_\phi$, $\Delta_{SS}$ and $\Delta_{ST}$.

## 2.1 Ideas about separating $\mathcal{C}_+$ not used in this work

*Remark* 2.1. Another way to distinguish $\mathcal{C}_+$ from $\mathcal{H}$ would be as follows. Primaries of $\mathcal{H}$ transform in irreps of $O(2N)$. Decomposing these irreps under $O(N) \times O(2)$, each primary of $\mathcal{H}$

---

[10] Just like $\phi^3$ in the Wilson-Fisher fixed point is a descendant of $\phi$ [51].

[11] As usual, we denote by $\mathcal{O}', \mathcal{O}'', \ldots$ the subsequent primaries having the same quantum numbers as $\mathcal{O}$.

Table 1: Operators treated in our study as isolated; ext.=external, int.=internal.

| name | $\ell$ | $\mathcal{R}$ | $\Delta$ | Note |
|---|---|---|---|---|
| $\phi$ | 0 | VV | | ext./int. |
| $s$ | 0 | SS | | ext./int. |
| $T_{\mu\nu}$ | 2 | SS | $d$ | int. |
| $J_\mu^{O(N)}$ | 1 | AS | $d-1$ | int. |
| $J_\mu^{O(2)}$ | 1 | SA | $d-1$ | int. |
| $\mathcal{O}_{ST}$ | 0 | ST | | int. |

gives rise to several primaries in different $O(N) \times O(2)$ irreps having exactly the same scaling dimension. These exact degeneracies are not expected in $\mathcal{C}_+$. For example, the conserved currents of $O(2N)$ would give, upon reduction under $O(N) \times O(2)$, conserved spin-1 operators in the AT and TA irreps. On the other hand, conserved currents in these irreps are not expected in $\mathcal{C}_+$. We could have imposed small gaps above the unitarity bound in the AT and TA spin-1 channels, with the goal of excluding $\mathcal{H}$ and keeping $\mathcal{C}_+$. In this work we will not use these gaps, because instead of excluding $\mathcal{H}$, we will be able to isolate $\mathcal{H}$ and $\mathcal{C}_+$ into two separate islands, thanks to the ST channel gap assumption.

*Remark* 2.2. Here is one more idea which we cannot rely on in this work, but which could be useful in future studies.[12] We know that $\mathcal{C}_+$ is stable and $\mathcal{H}$ is unstable, hence $\mathcal{H}$ contains one more relevant SS scalar. This operator can be written as $\mathcal{O} = h_{IJKL} W^{IJKL}$ where $W$ is a rank-4 symmetric traceless primary of $O(2N)$ and $h_{IJKL}$ is a tensor which breaks $O(2N)$ to $O(N) \times O(2)$. Can we use the existence of $\mathcal{O}$ to distinguish $\mathcal{H}$ from $\mathcal{C}_+$? Unfortunately, in our setup this will not work, because the $O(2N)$ selection rules of $\mathcal{H}$ preclude the appearance of $\mathcal{O}$ in the OPEs $\phi \times \phi$, $s \times s$ to which we are sensitive. To be sensitive to $\mathcal{O}$, we would have to enlarge the setup by including external fields in $T$ representations of $O(N)$ and/or $O(2)$. This is left for the future.

## 2.2 Unitarity assumption

The unitarity assumption about the CFT is going to play an important role in our numerical conformal bootstrap analysis. Namely we will be assuming, as usual, reality of scaling dimensions $\Delta_i$ and of the OPE coefficients $f_{ijk}$, and the unitarity bounds on $\Delta_i$. Strictly speaking, these constraints are satisfied only for integer $N$ and $d$. Indeed it is known that CFTs for non-integer $N$ [52] and non-integer $d$ [53,54] are not unitary. However, one can argue [54] that violations of unitarity for $d \geqslant 2$ are restricted to the sector of high-dimension operators, whose exchanges contribute exponentially little to CFT four-point functions of low-dimension operators that we will be studying. At present these effects are likely below the precision of numerical conformal bootstrap algorithms. In agreement with this intuition, the Ising model for $2 \leqslant d \leqslant 4$ was studied in [55–58] under the unitarity assumption, strictly speaking invalid for non-integer $d$, finding no inconsistency. On the other hand going below $d = 2$ without accounting for the loss of unitarity did lead to suspect results [59].

Analogously, one can argue that violations of unitarity should be negligible in the $O(N)$ model provided that $N$ is sufficiently large. A rule of thumb is to require that the dimensions of the exchanged $O(N)$ representations should be non-negative. E.g. the antisymmetric two-index tensor has dimension $N(N-1)/2$, so if it is exchanged, we should impose $N \geqslant 1$. This intuition is confirmed by the numerical conformal bootstrap analysis of the $O(N)$ CFT in $3 \leqslant d \leqslant 4$, $1 \leqslant N \leqslant 3$ [60]. There, assuming unitarity, the $O(N)$ CFT was isolated to a

---

[12]We thank Ning Su for discussions concerning this remark.

bootstrap island whose position was in excellent agreement with the resummed $\epsilon$-expansion. Furthermore, this island disappeared when $N \to 1^+$. The conclusion is that non-unitarity was negligible for $N \geqslant 1$ but not for $N < 1$. Indeed, a low-lying state CFT state whose norm becomes zero at $N = 1$, and would have become negative for $N < 1$, was identified in [60].

As already mentioned, the $O(N) \times O(2)$ fixed point $\mathcal{C}_+$ is expected to merge and annihilate with $\mathcal{C}_-$ at a critical value $N_c(d)$. Our working assumption will be that violations of unitarity are negligible for $N > N_c(d)$, when the fixed points $\mathcal{C}_\pm$ have real couplings. For $N < N_c(d)$ the fixed points go into the complex plane and continue their life there as "complex CFTs" [61, 62], having complex scaling dimensions and complex OPE coefficients, which is a much more violent violation of unitarity. Thus, while we expect to find solutions to the unitary bootstrap equations at $N > N_c(d)$, we may hope that these solutions will disappear at $N < N_c(d)$. This will be our method of determining $N_c(d)$.

## 2.3 Gap assumptions

The gap in the symmetry channel XY of spin $\ell$ will be denoted $\mathtt{gapXY}_\ell$. This means that all operators in this channel, except for the isolated operators from Table 1, are assumed to satisfy $\Delta \geqslant \mathtt{gapXY}_\ell$. The first group of gap assumptions is:

$$\mathtt{gapSS}_0 = d\,, \tag{7a}$$

$$\mathtt{gapSS}_2 = d + 1\,, \tag{7b}$$

$$\mathtt{gapAS}_1 = (d-1) + 1\,, \tag{7c}$$

$$\mathtt{gapSA}_1 = (d-1) + 1\,. \tag{7d}$$

These are easy to motivate. The $\mathrm{SS}_0$ assumption is simply the fixed point stability condition of $\mathcal{C}_+$ (see as well Remark 2.2). In the $\mathrm{SS}_2$, $\mathrm{AS}_1$ and $\mathrm{SA}_1$ channels, we have respectively $T_{\mu\nu}$ of dimension $d$, and the conserved currents of dimension $d-1$. At large $N$, the gaps above these operators are equal to 2 at leading order. The increment 1 in (7b), (7c), (7d) was chosen in order to be comfortably below this large $N$ value.

Let us discuss next the $\mathrm{VV}_0$ channel, which has two relevant operators $\phi$ and $\phi'$. The operator $\phi$ is treated as isolated. The leading order predictions for the dimension of $\phi'$ in the $\epsilon$-expansion and at large $N$ are:

$$\Delta_{\phi'} = \begin{cases} 3 \times \frac{d-2}{2} + O(\epsilon)\,, \\ \frac{d-2}{2} + 2 + O(1/N)\,. \end{cases} \tag{8}$$

We are not aware of results beyond the leading order. To be safe, we will use values for $\mathtt{gapVV}_0$ comfortably below the $\epsilon$-expansion result from (8) which is the smaller prediction:

$$\mathtt{gapVV}_0 = d - 2\,. \tag{9}$$

As already mentioned, we will also need a gap assumption in the $\mathrm{ST}_0$ channel. The precise value of $\mathtt{gapST}_0$ will be informed by the value of $\Delta_{\mathrm{ST}'}$ in $\mathcal{H}$ and $\mathcal{C}_+$, given in Eqs. (D.3),(D.8) to the first subleading order in $1/N$. In Fig. 4 we plot these values as a function of $d$, for the value $N = N_c^\epsilon(d)$, which is the $\epsilon$-expansion for $N_c(d)$, discussed in Appendix E.1. Given these curves, we will consider

$$\mathtt{gapST}_0 = \frac{d}{2} + \frac{3}{2}\,, \tag{10}$$

a conservative gap assumption, since it lies comfortably below the large-$N$ predictions for $\Delta_{\mathrm{ST}'}$ in both $\mathcal{C}_+$ and $\mathcal{H}$. In Section 3.4, we will also consider the effect of varying $\mathtt{gapST}_0$.

In all other channels, $\mathtt{gapXY}_\ell$ will be set to the unitarity bound.

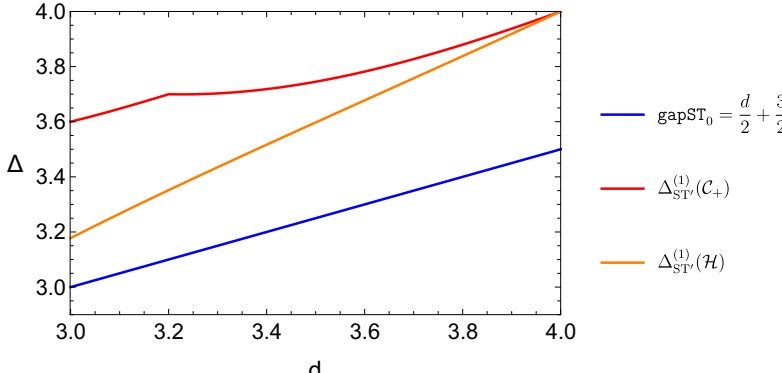

Figure 4: Comparison of the gap assumption $\texttt{gapST}_0 = \frac{d}{2} + \frac{3}{2}$ with the dimension of $\mathcal{O}_{\text{ST}'}$ in $\mathcal{H}$ and $\mathcal{C}_+$ at the first subleading order in $1/N$, evaluated at $N = N_c^\epsilon(d)$ from the $\epsilon$-expansion. The $\mathcal{C}_+$ curve has a kink due to crossing between the $\gamma_1$ and $\gamma_2$ coefficients from (D.7).

## 3  Conformal bootstrap analysis

As we saw in the previous section, our setup leads to a three-dimensional parameter space $\mathcal{P}$ consisting of points $x = (\Delta_\phi, \Delta_{\text{SS}}, \Delta_{\text{ST}})$, which are respectively the dimensions of the first primary scalars in the VV, SS and ST channels. With our gap assumptions not every point in this space is allowed, and our first order of business will be to get an idea of the allowed regions for various $N$ and $d$.

We will search for allowed points by the navigator function method [49], which was already used in several bootstrap studies [58, 60, 63–65]. We recall that the navigator function is constructed so that it is positive on disallowed points and negative on allowed points. Therefore, to find allowed points we can use a local minimization algorithm for the navigator function. Technical details of the navigator approach are discussed in Appendix F. Some parts of our study also used the new $\texttt{skydive}$ algorithm [50] for increased efficiency, which solves both the navigator optimization problem and the semi-definite programming problem at the same time.

We will first use the navigator method to find islands in the three-dimensional space $\mathcal{P}$ for fixed $N, d$. If by varying $N$ or $d$ an island disappears, or equivalently a negative local minimum turns positive, then this is evidence that the corresponding CFT ceases to exist. We will determine the exact point at which this happens by including $N$ in our search, so we will 'navigate' towards the vanishing point of the island in the four-dimensional space spanned by $x$ and $N$. Our notation below will reflect these two different setups: although the navigator function $\mathcal{N}$ always depends on all five variables $N, d$, and $x = (\Delta_\phi, \Delta_{\text{SS}}, \Delta_{\text{ST}})$ we will suppress the directions that we hold fixed and write $\mathcal{N}(x)$ if we move in three dimensions and $\mathcal{N}(x, N)$ if we move in four.

In all our searches we will hold fixed the gap assumptions of the previous subsection.

### 3.1  Finding islands for large $N$

We will begin by fixing $N$ to be significantly larger than the expected critical value. In this region the existence of chiral fixed points is not in doubt, and our first task will be to show that the conformal bootstrap analysis is able to isolate them. We will carry out the analysis first near $d = 4$, namely for $d = 3.8$ and $N = 20$. We will then carry out similar analysis for $d = 3$ and $N = 8$. With $d$ and $N$ fixed we will navigate in the remaining three variables $x = (\Delta_\phi, \Delta_{\text{SS}}, \Delta_{\text{ST}})$ and look for islands of allowed points.

**SciPost**

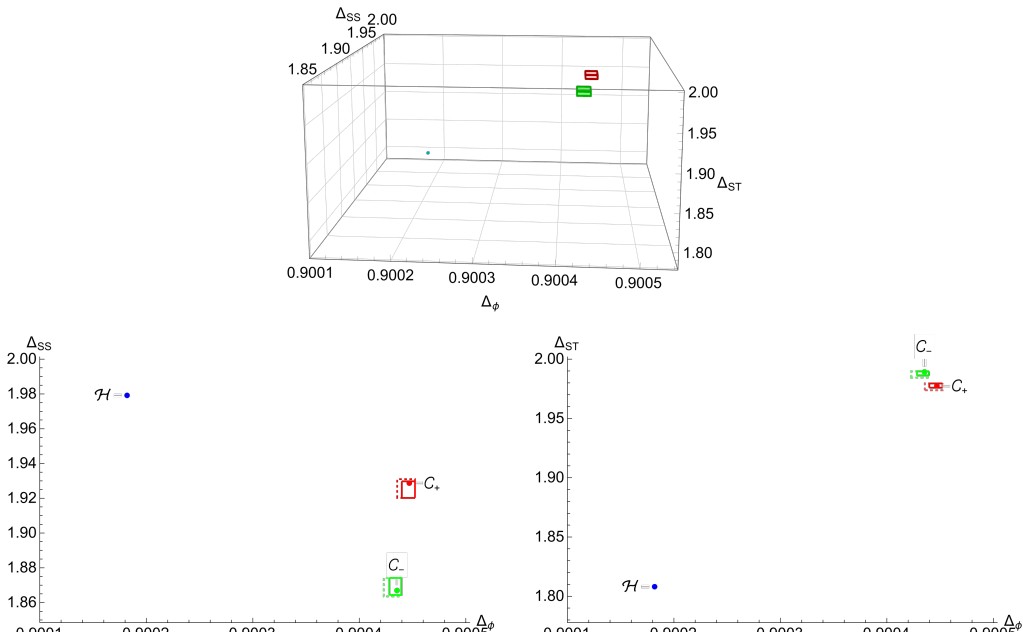

Figure 5: *Top:* Minimal bounding boxes containing three isolated allowed regions, at $\Lambda = 31$, in the space of $O(N) \times O(2)$-symmetric unitary[14] CFTs at $N = 20$, $d = 3.8$ under the gap assumptions (7), (9), (10). *Bottom:* The solid lines are the projections of these allowed regions to the planes of $(\Delta_\phi, \Delta_{SS})$ and $(\Delta_\phi, \Delta_{ST})$. The dashed lines indicate the same bounds if weakening (10) to $\texttt{gapST}_0 = \frac{d}{2}+1$. For reference, we also include the perturbative estimates of the $\epsilon$-expansion for the locations of $\mathcal{H}$ (blue), $\mathcal{C}_+$ (red) and $\mathcal{C}_-$ (green), see Appendix E.2. An allowed region containing the fixed point $\mathcal{C}_-$ exists at this $\Lambda$ despite the actual physical fixed point violating the stability assumption (7a), indicating insufficient sensitivity to this constraint. We checked that all three islands still survive at the higher derivative order $\Lambda = 43$ (see footnote 15).

### 3.1.1 $d = 3.8$ and $N = 20$

We begin our search by exploring $d = 3.8$ and $N = 20$. The former is close to $d = 4$, so we will be able to match our numerical results to the $\epsilon$-expansion, and the latter is moderately above the $\epsilon$-expansion prediction $N_c^\epsilon(3.8) = 17.3997(6)$ (see Appendix E.1) so we expect $\mathcal{C}_+$ to exist. The result of our investigations at the derivative order $\Lambda = 31$ (see Appendix F for the numerical details) is shown in Fig. 5. Within the shown ranges we find that almost every point is excluded with the exception of three small allowed regions in close vicinity of the perturbative predictions for the scaling dimensions of $\mathcal{H}$, $\mathcal{C}_+$ and $\mathcal{C}_-$.[13]

Before we discuss the physical consequences of this result, let us explain the precise meaning of the three boxes shown in Fig. 5. For each region we first found a point where the navigator function was negative, indicating that the point was allowed. By continuity this point should be part of a larger allowed region, called an "island", which one could in principle delineate by carefully tracing its boundary where the navigator function is exactly zero. To save computational time, we have however chosen not to do so, and instead simply determined the extremal dimensions of the islands along the three axes of the search space $\mathcal{P}$. The islands are thus contained within the boxes shown in Fig. 5, or as we say "boxed in." See Appendix F for further details.

---

[13]A further allowed region exists for larger $\Delta_\phi$, outside the range shown in the plot. This region, known as the "peninsula", is analogous to the one first identified in the three-correlator study for the 3D Ising model [34].

[14]See the caveats discussed in Section 2.2.

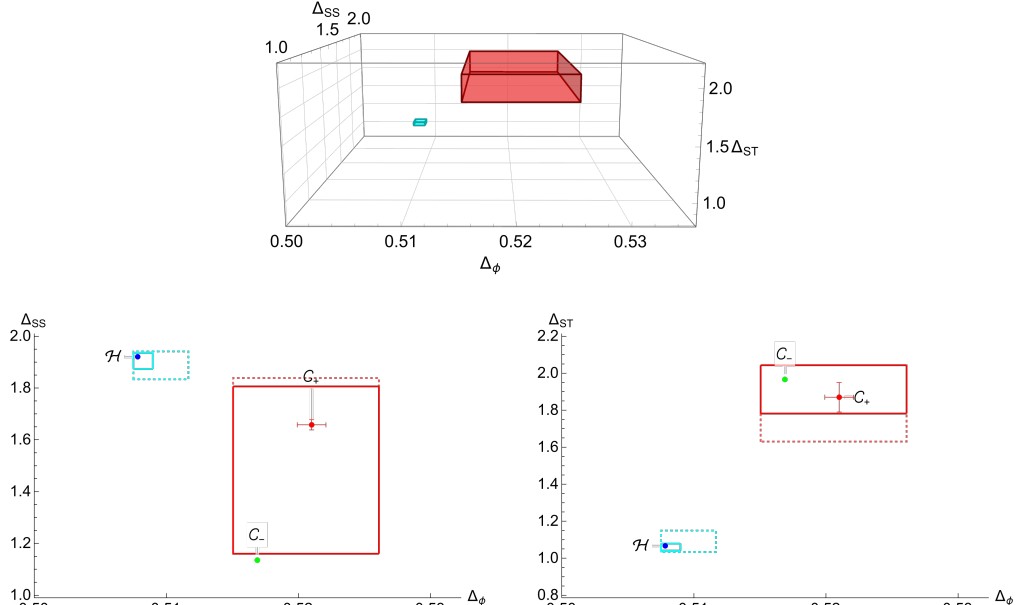

Figure 6: *Top:* Minimal bounding boxes containing isolated allowed regions in the space of $O(N) \times O(2)$ symmetric unitary CFTs for $N = 8$, $d = 3$ under the assumptions (7), (9), (10) at $\Lambda = 31$. *Bottom:* Projections of these allowed regions on the $(\Delta_\phi, \Delta_{SS})$ and $(\Delta_\phi, \Delta_{ST})$ planes (solid). The dashed lines indicate the same bounds under the weaker gap assumption $\mathtt{gapST}_0 = \frac{d}{2} + 1$. For reference, we also include the resummed field theory estimates of [6, 66], along with their uncertainty, for $\mathcal{C}_+$; and the unresummed large-$N$ expansion for $\mathcal{H}$ and $\mathcal{C}_-$ (see the end of Appendix E.2). In this plot we did not find a separate isolated region associated to $\mathcal{C}_-$.

The most encouraging aspect of Fig. 5 is the clean separation between the different fixed points. To see why, suppose that we had instead found one big allowed region encompassing all three theories. For lower $N$ we would then know that this big region can never disappear, simply because $\mathcal{H}$ always exists, which would make it impossible to determine $N_c$. The possible way out of that impasse could have been to exclude $\mathcal{H}$ from the analysis by making stronger gap assumptions, such as increasing $\mathtt{gapST}_0$. But this would have been very delicate since the estimated values for this gap in $\mathcal{C}_+$ and $\mathcal{H}$ are close (see Fig. 4). Fortunately the separated islands that we found, shown in Fig. 5, instead allow us to proceed differently: we can simply determine $N_c(d)$ by the vanishing of the $\mathcal{C}_+$ and $\mathcal{C}_-$ islands at lower $N$, all without worrying about the continued existence of the $\mathcal{H}$ island. This is how we will proceed below.

Finally we should comment on the unexpected appearance of the $\mathcal{C}_-$ island. The $\mathcal{C}_-$ fixed point is RG-unstable, i.e. it has a second relevant singlet scalar, and in principle it should be ruled out by the gap assumption (7a). That we still see an allowed region around $\mathcal{C}_-$ means that our bootstrap setup is not sufficiently sensitive to this particular gap assumption (although it is sensitive to other assumptions, because the allowed region is actually quite small). This may be due to the additional singlet scalar being only slightly relevant, since $N = 20$ is not too far above $N_c^\epsilon(3.8)$. We checked that the $\mathcal{C}_-$ island, as the other two, survives also at $\Lambda = 43$.[15]

---

[15]Namely, we checked that the $\Lambda = 43$ navigator is negative at the minima of the $\Lambda = 31$ navigator inside each of the three islands.

### 3.1.2 $d = 3$ and $N = 8$

In Fig. 6, we show the results of the same exercise carried out for $d = 3$ and $N = 8$, which is close to but above the $\epsilon$-expansion value $N_c^\epsilon(3) = 5.9(2)$ obtained as in Appendix E.1.[16] In this case we see two disconnected islands, one around $\mathcal{H}$, the other around $\mathcal{C}_+$. The islands are significantly larger than in Fig. 5. Predictions of the (resummed) $\epsilon$-expansion for the scaling dimensions at the $\mathcal{H}$ and $\mathcal{C}_+$ fixed points comfortably fall into the corresponding islands. In this figure, unlike for $d = 3.8$, there is no separate disconnected island around the $\mathcal{C}_-$ fixed point; instead, the predicted $\mathcal{C}_-$ scaling dimensions fall inside the same island as $\mathcal{C}_+$ (or close to it). Once again, this shows low sensitivity to the gap assumption (7a). While there is no separate $\mathcal{C}_-$ island, the navigator function has two separate minima inside the island containing both fixed points, close to their predicted positions, see Fig. 8 below.

### 3.2 $N_c(d)$ from the disappearance of the $\mathcal{C}_+$ island

In Section 3.1 we chose $N$ to lie well above its critical value predicted by the $\epsilon$-expansion. We showed bootstrap evidence that for those values of $N$ the $\mathcal{C}_\pm$ fixed points do exist, and have operator dimensions consistent with the $\epsilon$-expansion.

We will now lower $N$ and determine the critical $N_c$ at which the island containing $\mathcal{C}_+$ and $\mathcal{C}_-$ disappears. We will denote the corresponding conformal bootstrap determination as $N_c^{\text{CB}}(d)$. This $N_c^{\text{CB}}(d)$ should be seen as a rigorous lower bound on the true $N_c(d)$. It is a lower bound because the island can disappear only faster with improved numerical precision, and it is rigorous because all our gap assumptions were rather conservative.

As mentioned above, for this analysis we consider $N$ as a varying parameter of the navigator function, on par with the CFT data variables $x = (\Delta_\phi, \Delta_{\text{SS}}, \Delta_{\text{ST}})$. Thus, we will be considering four-dimensional allowed regions where the navigator function

$$\mathcal{N}(\Delta_\phi, \Delta_{\text{SS}}, \Delta_{\text{ST}}, N) \tag{11}$$

is negative. The $\mathcal{C}_+$ island now lives in the four-dimensional parameter space, and finding $N_c^{\text{CB}}(d)$ is equivalent to finding the extremal point of the island in the $N$ direction, i.e. the point where the navigator minimum turns from negative to positive. See Appendix F.5 for technical details. In this way we determined $N_c^{\text{CB}}(d)$ in the range $3 \leqslant d \leqslant 3.8$ in steps of $0.1$.[17]

The result of this analysis was shown in Fig. 3 on page 6. This plot, which directly addresses question (4), is the main result of our paper. In Fig. 3 we also show for comparison $N_c^\epsilon(d)$ from the $\epsilon$-expansion (Appendix E.1) and $N_c^{\text{NPRG}}(d)$ from the NPRG results of [3, Fig. 9]. For $d$ close to 4, the curve $N_c^\epsilon(d)$ should be trustworthy. In this region, our curve $N_c^{\text{CB}}(d)$ shows a rather good agreement with $N_c^\epsilon(d)$ to linear order in $\epsilon$, while $N_c^{\text{NPRG}}(d)$ shows a small but noticeable negative first-order deviation from $N_c^\epsilon(d)$. In the range $3.5 \leqslant d \leqslant 3.8$ the curve $N_c^{\text{CB}}(d)$, which as mentioned is a rigorous lower bound on $N_c$, lies above the NPRG curve and rules it out.

As $d$ approaches 3, the difference between the NPRG and $\epsilon$-expansion predictions decreases, and we have:

$$N_c^\epsilon(3) = 5.96(19), \qquad N_c^{\text{NPRG}}(3) = 5.1. \tag{12}$$

On the other hand our curve lies quite a bit lower than the other two in this range, and it ends in $d = 3$ at

$$N_c^{\text{CB}}(3) = 3.78. \tag{13}$$

---

[16]This number is close to $N_c(3) = 5.96(19)$ obtained in [6] from the $\epsilon$-expansion, which was obtained using a different method from the one described in Appendix E.1 and instead was based a combination of several resummation techniques.

[17]We note that there is a subtlety in our reasoning which has to do with the fate of the islands as we lower $N$ continuously from large values, where they are isolated, to $N_c$, where they disappear. This will be discussed below in Section 3.4.3.

Although this value is significantly lower than $\epsilon$ and NPRG values, we stress again that it is a rigorous lower bound. Importantly, even this lower bound (13) is strong enough to rule out the physically relevant values $N = 2, 3$.

A final feature of our curve is that it shows no sign of the turnaround behavior predicted to happen by the fixed-dimensional RG in Fig. 2 for $d \approx 3.2$.

We emphasize again that we used the gap assumptions laid out in Section 2.3, in particular (7), (9), as well as (10) for $\mathtt{gapST_0}$. As mentioned, we can afford putting this gap conservatively low, since our method does not require eliminating the $\mathcal{H}$ island. It is interesting to inquire how our $N_c(d)$ depends on the imposed gap assumptions, in particular on $\mathtt{gapST_0}$. This will be done in Section 3.4 below for $d = 3$, where we will see that the dependence on $\mathtt{gapST_0}$ is rather weak in a certain range around (10). On the other hand, we could try to improve $N_c^{\mathrm{CB}}(d)$ by increasing $\Lambda$ or by considering more correlators. These improvements will be left for the future.

## 3.3 Tracking the navigator minima

In Section 3.1 we have found isolated bootstrap islands for $N$ much above the critical value, and for two values of $d$. Being in the region of large $N$ and, for $d = 3.8$, also close to $d = 4$, the position of those islands could be with confidence identified with the Heisenberg and the chiral fixed points of the $O(N) \times O(2)$ model. Then, in Section 3.2 we found that bootstrap islands are disappearing along the curve $N = N_c^{\mathrm{CB}}(d)$ shown in Fig. 3. The gap assumptions which went into the determination of the islands are expected to be satisfied by both the Heisenberg and the chiral fixed points. The Heisenberg fixed point exists for any $N \geqslant 1$ and the island containing it is not expected to disappear as $N$ decreases. Naturally, we associated the disappearing islands with the chiral fixed point.

In this section we will provide further evidence confirming that the identification of the disappearing islands with the chiral fixed point is correct. For this we will track, for two values of $d$, the minimum of the navigator function with respect to $(\Delta_\phi, \Delta_{\mathrm{SS}}, \Delta_{\mathrm{ST}})$, within the $\mathcal{C}_+$ island, as a function of $N$. Varying $N$ from large values down to $N_c$, we will show that the minimum varies continuously. We track minima and not the whole islands, since it would be much more costly numerically to determine the island extensions in every $N$.

### 3.3.1 $d = 3.8$

The position of $\mathcal{C}_+$ navigator minima for $d = 3.8$ and for $N$ ranging from $N = 30$ down to $N_c^{\mathrm{CB}}(3.8)$ is shown in Fig. 7.[18] The leftmost points correspond to $N = N_c^{\mathrm{CB}}(3.8) = 17.1585$ which is where the $\mathcal{C}_+$ island disappears for our chosen value of $\Lambda = 31$. In the same plots we show by red curves the $\epsilon$-expansion prediction for the same quantities, computed as in Appendix E.2. These curves end at $N_c^\epsilon = 17.3997(6)$ for $d = 3.8$. We see that the navigator minimum varies continuously with $N$, tracking closely the $\epsilon$-expansion curve, except for a few leftmost points, close to $N_c$. We interpret the difference between $N_c^{\mathrm{CB}}$ and $N_c^\epsilon$, as well as the residual deviations between the bootstrap points and the $\epsilon$-expansion curves as numerical artifacts, expected to disappear as $\Lambda \to \infty$. We have indeed observed that these deviations decrease as $\Lambda$ is increased through $\Lambda = 19, 23, 27$ (not plotted) to $\Lambda = 31$.

These plots also allow us to demonstrate another expected feature of the mechanism behind the disappearance of the chiral fixed point at $N \to N_c(d)^+$. As mentioned, this should happen via "merger and annihilation": two fixed points $\mathcal{C}_\pm$ which exist at $N > N_c(d)$ merge

---

[18]The gap assumptions used were the same as in Fig. 5, except for the use of the aggressive $\mathtt{gapST_0} = 3.85$, roughly halfway between the $\mathcal{C}_+$ and $\mathcal{H}$ large-$N$ curves in Fig. 4 at $d = 3.8$. As $N$ grows, the $\mathcal{C}_+$ dimension of $\mathcal{O}_{\mathrm{ST}}$ tends to 4, so this gap should be valid for $N \geqslant N_c(3.8)$.

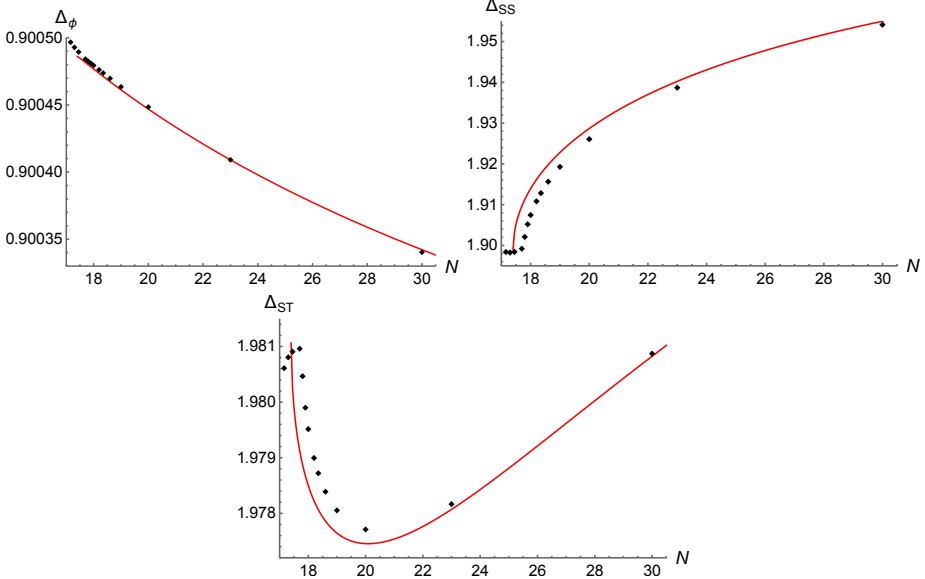

Figure 7: Diamonds: $\Delta_\phi, \Delta_{SS}, \Delta_{ST}$ at the minimal navigator point within the $\mathcal{C}_+$ island, as a function of $N$, for $d = 3.8$ and the derivative order $\Lambda = 31$. The leftmost points corresponds to $N = N_c^{CB}(3.8) = 17.1585$ which is the critical $N$ value at which the $\mathcal{C}_+$ island disappears. Red curves: the $\epsilon$-expansion prediction for the same quantities.

into a single CFT at $N = N_c(d)$. At even lower $N$, the fixed points $\mathcal{C}_\pm$ would have complex couplings and complex anomalous dimensions. These are "complex CFTs" [67], and they become invisible in our studies since we assume real scaling dimensions.

The merger and annihilation scenario was discussed in various contexts in many works, notably [65, 67–70]. Its telltale signature is a near-marginal singlet scalar operator $\mathcal{O}$ whose scaling dimension shows a square-root behavior near the merger point (see e.g. [67], Eq.(2.4))

$$\Delta_{\mathcal{O}} \approx d \pm const.\sqrt{N - N_c(d)} \qquad (N - N_c(d) \ll 1), \tag{14}$$

where the sign $\pm$ corresponds to $\mathcal{C}_\pm$. This behavior can be linked to a square-root singularity in the coupling of $\mathcal{O}$ near the merger.

For us $\mathcal{O} = SS'$, the second-lowest SS scalar. In our numerical study, we found it hard to extract its dimension reliably using the Extremal Functional Method [31], likely because it is too close to the gap assumption (7a). We leave direct numerical bootstrap verification of (14) to future work.

However, since related to a square-root singularity of a coupling, square-root behavior should be present not just in $\Delta_{\mathcal{O}}$ but in any scaling dimension or any OPE coefficient (see [68], or [62], Fig. 7). And indeed, in Fig. 7 we clearly see such a square-root behavior in $\Delta_{SS}$ and $\Delta_{ST}$ data. According to the $\epsilon$-expansion, the expected square-root singularity in these quantities is quite pronounced, so it is easy for us to detect it numerically. For the third quantity $\Delta_\phi$, the $\epsilon$-expansion predicts a tiny square-root singularity not visible on the scale of Fig. 7.

*Remark* 3.1. We note a recent conformal bootstrap attempt to observe the merger and annihilation scenario in a different model [65]. That paper studied the critical and tricritical point of the 3-state Potts model as a function of dimension $d \geqslant 2$. Merger and annihilation is supposed to occur at some $d = d_c$ between 2 and 3. As a sign of this, a square-root behavior of multiple scaling dimensions seems be setting in, as expected, in their Fig. 2, at least sufficiently below the merger point. However in their study this behavior is replaced closer to the merger point

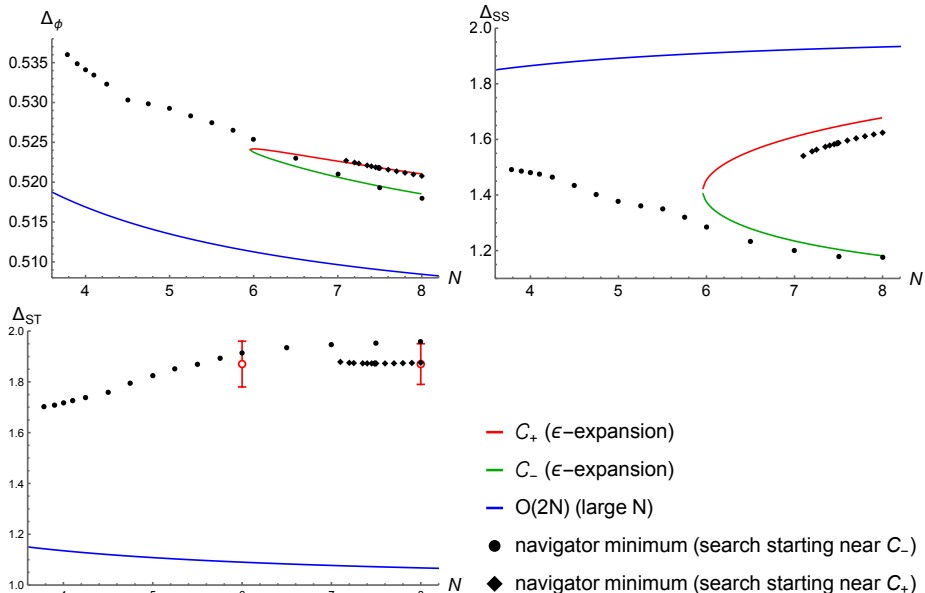

Figure 8: Diamonds and dots: positions of the $\mathcal{C}_+$ and $\mathcal{C}_-$ navigator minima for $d = 3$, as a function of $N$, at the derivative order $\Lambda = 31$. The leftmost points corresponds to $N = N_c^{\mathrm{CB}}(3) = 3.78$ where the $\mathcal{C}$ island disappears. Red, green, blue curves: the $\epsilon$- and $1/N$-expansion predictions. Red error bars: estimate of $\Delta_{\mathrm{ST}}$ in $\mathcal{C}_+$ from fixed-dimensional RG (see [66, Table III] and Appendix E.2).

by a linear approach, which is not expected theoretically and is likely a numerical artifact. Our $d = 3.8$ results in Fig. 7 show a clearer picture. Note that in our study we managed to isolate the $\mathcal{C}_+$ fixed point to an island, which was not the case for the theories studied in [65].

### 3.3.2 $d = 3$

We now move to $d = 3$ and show in Fig. 8 the position of navigator minima for $N$ ranging from $N = 8$, the value in Fig. 6, down to $N_c^{\mathrm{CB}}(3) = 3.78$ where the $\mathcal{C}$ island disappears.

We observe several different features compared to Fig. 7. First, at $N = 8$ the navigator function now has two well separated minima, close to the respective expected positions of $\mathcal{C}_\pm$ within the island. We track both minima which we therefore tentatively call the $\mathcal{C}_+$ navigator minimum (diamonds) and the $\mathcal{C}_-$ minimum (dots).

As we decrease $N$ we find that the $\mathcal{C}_+$ minimum is the first to disappear, around $N = 7$. At this point the local minimum collides with a saddle, leading to the disappearance of both. Precisely at this transition, one Hessian eigenvalue is zero, which we checked by explicitly computing the Hessian. However, the navigator function itself is still negative so we are within the island.[19]

The $\mathcal{C}_-$ minimum on the other hand, which we recall now lies within the same island, continues to exist for lower values of $N$. As shown in Fig. 8, it varies continuously with $N$ until it becomes a positive minimum at $N = N_c^{\mathrm{CB}}(3) = 3.78$, at which point the $\mathcal{C}$ island disappears.

Various colored curves in Fig. 8 show the $\epsilon$-expansion or $1/N$-expansion predictions for the positions of various theories. The blue lines show the position of the $O(2N)$ fixed point. We

---

[19]While it is very interesting to note that all local minima of the navigator function discovered up to now seem to have their origin in physical theories it is important to keep in mind that the absence of the $\mathcal{C}_+$ minimum below $N < 7$ does not imply this theory does not exist. What is more important is that the conformal data values that such a stable CFT would have are not excluded by our current analysis for $N > 3.78$, but are rigorously excluded for $N < 3.78$.

see that the $\mathcal{C}_-$ minimum remains well separated from this line for all $N$ of interest. The red and green lines show the positions of the $\mathcal{C}_\pm$ fixed points, which according to the $\epsilon$-expansion annihilate at $N = 5.96(16)$. We see that the $\mathcal{C}_-$ minimum tracks closely the green $\epsilon$-expansion curve for $N \gtrsim 6.5$.

Fig. 8 in $d = 3$ paints a somewhat less satisfactory picture than Fig. 7 in $d = 3.8$. Ideally we would have found two persistent local minima, one positive (so excluded, corresponding to $\mathcal{C}_-$) and the other negative (corresponding to $\mathcal{C}_+$), which upon decreasing $N$ would merge precisely at $N_c^{\text{CB}}(3)$ and then remain positive for smaller $N$. Such a scenario would then naturally also produce the desired square-root behavior of the scaling dimensions.

Instead we see in Fig. 8 that the $\mathcal{C}_+$ minimum disappears too soon, followed by a rather large region between $N_c^{\text{CB}}(3) = 3.78$ and $N \approx 6.5$ without good agreement with the $\epsilon$-expansion. All of this could however be numerical artifacts, and in future work it would be interesting to see how the Fig. 8 evolves when increasing $\Lambda$ and whether the features at small $N$ remain or the bounds converge closer to the ideal scenario.[20]

On the positive side, Fig. 8 does demonstrate a continuous connection between the disappearing island at $N = N_c^{\text{CB}}(3)$ and the island at $N = 8$ which could be unambiguously associated with the $\mathcal{C}_\pm$ fixed points. It is therefore not in contradiction with our claim that $N_c^{\text{CB}}(3)$ is a rigorous lower bound for the disappearance of $\mathcal{C}_+$.

## 3.4 How does $N_c^{\text{CB}}(3)$ depend on the choice of $\text{gapST}_0$?

In this subsection we will investigate the dependence of $N_c^{\text{CB}}(d)$ on $\text{gapST}_0$. The best possible scenario would be that $N_c^{\text{CB}}(d)$ does not change at all if we vary $\text{gapST}_0$ within a certain interval. Of course we cannot increase $\text{gapST}_0$ above the actual dimension $\Delta_{\text{ST}'}$ of the second scalar in the ST representation since this would exclude the $\mathcal{C}_+$ fixed point altogether. We probably cannot decrease $\text{gapST}_0$ too much either, because then the allowed regions can become so big that the $\mathcal{C}_+$ island merges with either the $\mathcal{H}$ island or the larger peninsula region discussed in footnote 13.

The actual variation of $N_c^{\text{CB}}(3)$ with $\text{gapST}_0$ is shown in Fig. 9. Most importantly it shows a rather mild dependence as long as we choose $\text{gapST}_0$ between about 2.8 and 3.4, confirming the robustness of our estimate Eq. (13) in the sense that it does not depend significantly on this gap assumption. Note that the upper value of about 3.4 is roughly compatible with the first-order large $N$ estimate for $\Delta_{\text{ST}'}$ which is shown as the red dashed curve in the figure. We unfortunately see no evidence for a flattening of the curve with increasing $\Lambda$, but at the same time it is clear that our result is far from being converged.

Another way to check for the sensitivity with respect to $\text{gapST}_0$ is to extract the operator spectrum from the extremal functional at $N_c(3)$, which we did for the blue points in Fig. 9. In an ideal scenario there is simply no operator with a scaling dimension equal to $\text{gapST}_0$, as this would mean that we could vary $\text{gapST}_0$ a bit without changing the functional and the bound. It is however often the case that numerical artefacts spoil this scenario, and this appears to be the case here. Indeed, as we show in Fig. 10, the scalar spectrum in the ST sector consistently shows an operator at the gap for each value of $\text{gapST}_0$. We also verified that its OPE coefficient is non-vanishing: $\lambda_{\phi\phi\mathcal{O}_{\text{ST}'}}$ varies smoothly from 0.06 to 0.03 in the plotted window. (For comparison, we obtained $\lambda_{\phi\phi\mathcal{O}_{\text{ST}}} \approx 0.3$ and $\lambda_{\phi\phi\mathcal{O}_{\text{ST}''}} \approx 0.005$.) It is the square of these coefficients that multiplies the conformal blocks, whose normalization we took to be that of [44].) In future work it will be important to verify that both Fig. 9 flattens out and that $\lambda_{\phi\phi\mathcal{O}_{\text{ST}'}}$ decreases upon increasing $\Lambda$.

---

[20]At this point it is for example unclear whether there is any physical significance to the change of slopes in the conformal data that is observed around $N = 4.2$. This would be interesting to investigate further.

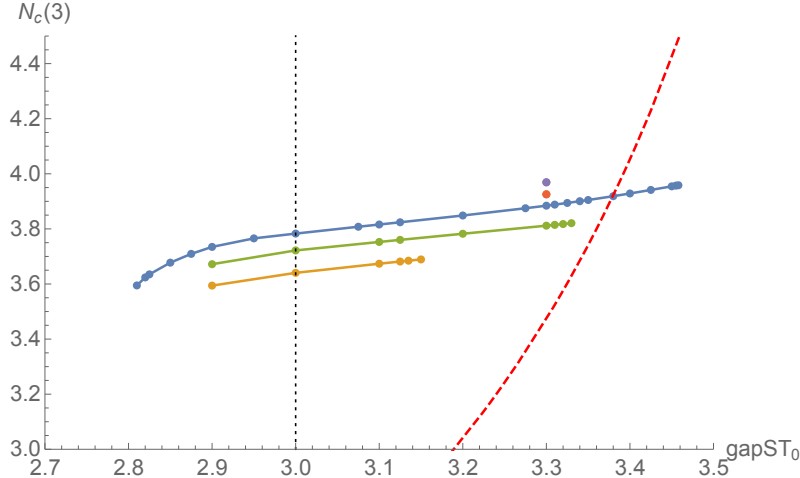

Figure 9: Lower bound on $N_c$ as found by the numerical conformal bootstrap as a function of the assumed value of $\texttt{gapST}_0$ at $\Lambda = 31$ (in blue). Some additional points are shown for $\Lambda = 27, 29, 33, 35$ in respectively orange, green, red, purple. The dotted vertical line corresponds to the previously chosen value $\texttt{gapST}_0 = 3$, yielding $N_c^{\text{CB}}(3) = 3.78$ at $\Lambda = 31$ as stated previously in Eq. (13). The dashed curved line indicates the large $N$ estimate of $\Delta_{\text{ST}'}$ in $\mathcal{C}_+$ (see Eq. (D.8)). It is natural to expect the $\mathcal{C}_+$ island to disappear somewhat to the right this curve.

In the remainder of this subsection we discuss interesting subtleties that arise from studying the endpoints of the $\Lambda = 31$ curve in Fig. 9. These endpoints came about as follows. Fig. 9 was obtained by simply repeating the analysis of Section 3.2 for different values of the $\texttt{gapST}_0$, using again the $\texttt{skydive}$ algorithm of [50], see Appendix F. In practice this procedure turned out to be very delicate for small and large values of $\texttt{gapST}_0$. We frequently observed runaway behavior, forcing us to restart with a slightly different initial point. The endpoints of the $\Lambda = 31$ curve are the last points for which we were able to obtain convergence.

### 3.4.1 Hessian and derivative information

Let us first discuss the behavior of the navigator function $\mathcal{N}(\Delta_\phi, \Delta_{\text{SS}}, \Delta_{\text{ST}}, N)$ in the vicinity of $N_c$ for a generic value of $\texttt{gapST}_0$. Recall that $N_c$ is the lowest point in the $N$ direction where

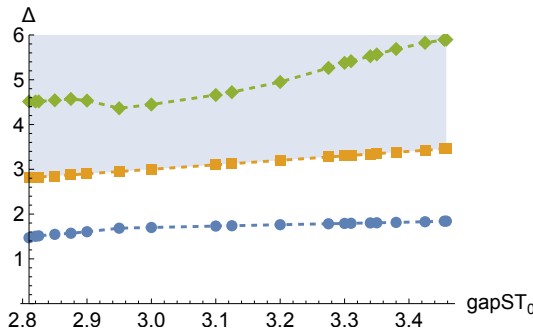

Figure 10: The first three scalar operators in the ST sector in the extremal spectrum as a function of $\texttt{gapST}_0$. (More precisely, this is the extremal spectrum corresponding to the blue points with $\Lambda = 31$ in Fig. 9.) The first operator ($\Delta_{\text{ST}}$, blue dots) is isolated, the second one ($\Delta_{\text{ST}'}$, orange dots) is precisely at the gap imposed by $\texttt{gapST}_0$ (as indicated by the blue area), and the third operator ($\Delta_{\text{ST}''}$, green dots) lies above it.

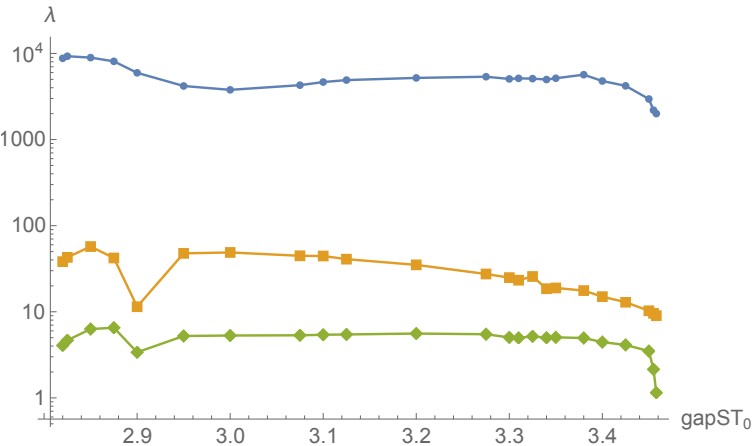

Figure 11: Eigenvalues of the Hessian $H$ of the navigator function at $N_c$ in the three $\Delta$ directions, as a function of $\texttt{gapST}_0$ for $d = 3$.

$\mathcal{N} \leqslant 0$. Since the navigator function is smooth, this must mean that:

$$
\begin{aligned}
\mathcal{N}(\Delta_\phi, \Delta_{SS}, \Delta_{ST}, N_c) &= 0, \\
\frac{\partial}{\partial N}\mathcal{N}(\Delta_\phi, \Delta_{SS}, \Delta_{ST}, N_c) &\leqslant 0,
\end{aligned}
\tag{15}
$$

where saturation of the inequality on the last line would be atypical and would imply a constraint on the higher derivatives such that the $\mathcal{N}$ is positive for $N$ a little bit below $N_c$. Likewise the navigator function must be at a local minimum with respect to the other parameters, so if we define its gradient and Hessian in $\Delta$-space at the critical point $N_c$ as

$$
\begin{aligned}
g_i &:= \frac{\partial}{\partial \Delta_i}\mathcal{N}(\Delta_\phi, \Delta_{SS}, \Delta_{ST}, N_c), \qquad \Delta_i \in (\Delta_\phi, \Delta_{SS}, \Delta_{ST}), \\
H_{ij} &:= \frac{\partial^2}{\partial \Delta_i \partial \Delta_j}\mathcal{N}(\Delta_\phi, \Delta_{SS}, \Delta_{ST}, N_c),
\end{aligned}
\tag{16}
$$

then we know that

$$
g_i = 0, \qquad \text{and} \qquad H \succ 0. \tag{17}
$$

Note that the second derivative in the $N$ direction is not constrained.

Experimentally the observed behavior agrees with the previous discussion for all the points where our algorithm converges. The algorithm itself guarantees that $\mathcal{N} = 0$ and $g_i = 0$ at optimality. Properties of the other derivatives of the navigator are shown in Figs. 11 and 12. In the first we show the eigenvalues of $H$, which are indeed all positive. In the second figure we include the $N$ direction. We first of all see that the gradient is negative, as expected. We also see that including the fourth direction yields a second-order derivative that is generally positive but turns negative at both endpoints.

### 3.4.2 Left endpoint behavior

At the leftmost point we observe from Fig. 12 that the gradient in the $N$ direction rapidly decreases in magnitude and that the four-dimensional Hessian has a negative eigenvalue. Since we have both gradient and Hessian information, we can construct a quadratic approximation to the real navigator function. This approximation predicts that $\mathcal{N}$ becomes negative *again* for $N$ somewhere below $N_c$. For example, for the leftmost point in Fig. 9 at $\texttt{gapST}_0 = 2.82$,

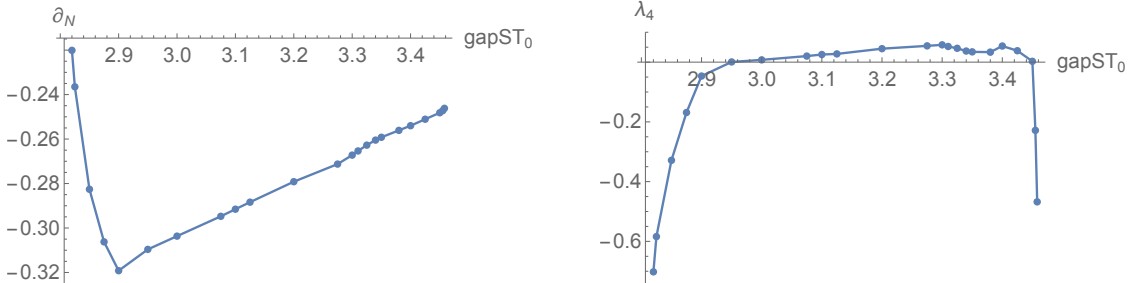

Figure 12: Behavior of the navigator function at $N_c$ in the $N$ direction, as a function of gapST$_0$ for $d = 3$. On the left we plot the gradient. On the right we plot the fourth eigenvalue of the Hessian of the four-dimensional navigator $\mathcal{N}(\Delta_\phi, \Delta_{SS}, \Delta_{ST}, N)$. We note that its corresponding eigenvector mostly in the $N$ direction, and the other three eigenvalues agree to within 10% with those shown in Fig. 11. This eigenvalue therefore accurately captures the second-order behavior of the navigator when one includes the $N$ direction. (We could have opted to simply plot the second partial derivative in the $N$ direction, but this always comes out positive and therefore does not convey the right information.)

where we obtained $N_c \approx 3.62$, this quadratic approximation would predict that the navigator becomes negative again at $N \approx 3.13$.

The analysis of gradient and Hessian therefore provides evidence for another allowed region that exists for $N < N_c$ and low values of gapST$_0$. This is not in disagreement with the expectations we formulated above, where we stated that gapST$_0$ must be chosen sufficiently large for the $\mathcal{C}_+$ island to (a) exist as an isolated island, and then (b) to disappear. It is however surprising that this additional region is disconnected, since it would have been more natural to see a smooth merging of the $\mathcal{C}_+$ island with another allowed region. Such a scenario would correspond to the three-dimensional $H$ becoming singular, but Fig. 11 shows no indication of such behavior near the left endpoint at low gapST$_0$.

In future work it will be important to investigate in detail the existence of a disconnected region below $N_c$. Is this region connected to the $\mathcal{H}$ island or the pensinsula? And is it due to the unnecessarily low value of gapST$_0$ or does it also persist for higher values? And is it related to the existence of fixed points below $N = 2$ predicted by the $\epsilon$ expansion as discussed in footnote 22?

### 3.4.3 Right endpoint behavior

The behavior at the right endpoint is more surprising. Here the main novel characteristic is the steep drop in the lowest eigenvalue of the three-dimensional $H$ as shown in Fig. 11. Naive extrapolation then predicts that the Hessian $H$ would become singular for a slightly higher value of gapST$_0$. We would then necessarily have a direction in $\Delta$ space in which the navigator decreases, simply because the leading term in that direction is the cubic term. The $\mathcal{C}_+$ island is then no longer isolated and instead gets connected to another region which does not disappear for $N$ below $N_c$. We do not know what this additional allowed region is, but the natural candidate would be the aforementioned peninsula that we know exists for all $N$.

Although we discovered this behavior at large gapST$_0$, it must actually also persist at lower gapST$_0$ because lowering a gap can only make the allowed region larger (at fixed $N$).

Let us therefore return to our earlier chosen value of gapST$_0 = 3$. If the above scenario is correct then the behavior is as follows. For $N = 8$ Fig. 6 shows an isolated $\mathcal{C}_+$ island. For $N$ slightly above $N_c$ there is likewise an isolated island (by continuity of the navigator function

and the positivity of $H$). But for intermediate values of $N$ there may not be a well-defined island. Instead, we would predict a merger with the peninsula and then again a re-detachment of an island.[21]

We emphasize that we observed this issue only for larger values of $\mathtt{gapST}_0$ simply because that is the only place where we investigated these intermediate values of $N$. Investigating the structure of the allowed regions in detail should be a priority for future numerical bootstrap studies.

# 4 Review of prior bootstrap studies

Let us review the previous bootstrap studies on the O($N$)×O(2) model [24, 25]. Comparison to our work will be done in the conclusions sections.

Ref. [24] was the first bootstrap work on the O($N$)×O(2) model. It was a single-correlator study, using only the 4pt function $\langle\phi\phi\phi\phi\rangle$. Working in $d = 3$ and specializing to $N = 3$, they found a mild kink in the gap maximization plot in the $\mathrm{ST}_{\ell=0}$ channel, as a function of $\Delta_\phi$. They observed that at the kink, $\Delta_\phi$ as well as the dimensions of scalars in the SS, ST, TS, TT, and AA representations extracted using the extremal functional method agree reasonably well with the values of these dimensions predicted for the putative chiral O(3)×O(2) fixed point of focus type found using the fixed-dimension expansion (for one of two RG schemes). Of course the focus fixed point, if it existed, would not be unitary since it would have a scaling operator with complex dimension. Ref. [24] assumed unitarity, and in particular real scaling dimensions for all operators. Be that as it may, Ref. [24] interpreted the existence of the kink and the above-mentioned agreement as evidence for the actual existence of a chiral O(3)×O(2) unitary fixed point.[22]

The more recent study [25] developed the initial findings of [24] in two directions:

1. With the same single-correlator setup as in [24], they carried out gap maximization in the $\mathrm{ST}_{\ell=0}$ and $\mathrm{TS}_{\ell=0}$ channels[23] for $N = 3, 4, 5, 10, 20$. They found that for large $N$ ($N = 10, 20$) when the corresponding unitary CFTs are reliably known to exist, there are kinks in these plots which match closely with the chiral ($\mathrm{TS}_{\ell=0}$) and antichiral ($\mathrm{ST}_{\ell=0}$) fixed points.[24] For smaller $N$ the kinks get milder, and their association with actual unitary CFTs is unclear.

Note the surprising inversion: at large $N$ the chiral fixed points are associated with $\mathrm{TS}_{\ell=0}$ gap maximization kinks [25], while for $N = 3$ they are associated with $\mathrm{ST}_{\ell=0}$ gap maximization kinks [24].

2. The second part of the study in [25] proceeds in the three-correlator setup $\langle\phi\phi\phi\phi\rangle$, $\langle\phi\phi ss\rangle$, $\langle ssss\rangle$ (i.e. the same as in our paper), imposing an assumption that the $\mathrm{TS}_{\ell=0}$ or $\mathrm{ST}_{\ell=0}$ gap saturates the corresponding gap maximization bound from the single-correlator analysis,

---

[21]An island that first grows and then shrinks as one decreases $N$ can also be found in the numerical investigations of the O($N$) models, see for example Fig. 1 of [36].

[22]It should be mentioned that Ref. [24] also discussed the "collinear" (also called "sinusoidal" [1]) fixed point, which has the quartic coupling $v < 0$. According to the state-of-the-art resummed $\epsilon$-expansion results, the collinear fixed point exists only when $N < 1.970(3)$ [6], by far excluding the physically interesting value $N = 4$ when such a fixed point, if it existed, could describe the chiral phase transition in QCD with two massless quark flavors. In this case, there are again fixed-dimension schemes which do find a collinear fixed point for $N = 3, 4$ [66, 71]. In the interest of space, and since our focus in this paper is on the chiral fixed point, we will not discuss this additional controversy here (see also [72]).

[23]Note that the symmetry group is O(2)×O($N$) in [25] while it is O($N$)×O(2) for us and [24]. Therefore the order of representations is reversed TS↔ST. We are using our conventions here. In [25], ST is called $W$ and TS is called $X$.

[24]This attribution goes back to [73], where O($N$) × O(3) CFTs were studied in $d = 3$, for $5 \leqslant N \leqslant 20$. There, the kinks in the $\mathrm{TS}_{\ell=0}$ and $\mathrm{ST}_{\ell=0}$ gap maximization plots at large $N$ were first identified with the chiral and antichiral fixed points. Refs. [24, 73] were the first bootstrap studies of multiscalar models with more complicated symmetry than O($N$).

plus several other reasonable gap assumptions on the spectrum. With these assumptions, they are able to turn kinks into islands of rather complicated shapes. For large $N$, these islands can be mapped to the chiral and antichiral CFTs.

Then they consider small $N = 3$, and for $\mathrm{ST}_{\ell=0}$ saturated gap (see the inversion noted above) the island they find agrees with the location of the putative O(3)×O(2) chiral focus-type fixed point (in the same $\overline{\mathrm{MS}}$ scheme that saw better agreement with the single-correlator results of [24]). Concerning the issue that the putative fixed point is focus-type, they say that "It is unclear to us how sizable nonunitarities could have been missed by our bootstrap results." As they point out, some previously found non-unitary CFT islands disappeared with increasing the constraining power of the numerics [74], and this may also happen to their island. They conclude by saying "Our results provided support for the existence of these fixed points, but we saw no signs of nonunitarity. Overall, we were unable to provide conclusive answers, but we believe that more dedicated bootstrap work with stronger numerics will be able to reach definitive conclusions in the near future."

# 5 Comparison to prior work, conclusions and outlook

Let us start by emphasizing the aspects in which our study which were different from prior work [24, 25], reviewed in Section 4. The setup of [24, 25], working in $d = 3$ and for a discrete sequence of $N$, could not clearly see if and how the putative O(3)×O(2) CFT connected to well-established O($N$)×O(2) CFTs at large $N$ and for $d$ close to 4. In contrast, our study investigated continuously the 2-dimensional parameter space $(d, N)$. We could therefore see the unitary O($N$)×O(2) CFTs disappear as $N$ approaches some critical curve $N_c(d)$ from above. The second difference between our study and [25] is in how the $\mathrm{ST}_{\ell=0}$ channel of the $\phi \times \phi$ OPE was treated in the multiple correlator setup. They set the first operator in this channel to saturate the single correlator bound, while making no assumptions about subsequent operators. We, on the other hand, treated $\Delta_{\mathrm{ST}}$ as a free parameter which was one of the arguments of the navigator function, while restricting the next operator in this channel to be above a gap gapST$_0$. We saw in Section 3.4 that with this assumption, $N_c^{\mathrm{CB}}(d = 3)$ shows only mild dependence on gapST$_0$, while remaining safely above 3. It is possible that the setup of [25] was simply not constraining enough to see a large-enough $N_c^{\mathrm{CB}}(d = 3)$. Finally, our study used $\Lambda = 31$, while [25] used $\Lambda \leqslant 25$.[25]

Our curve $N_c^{\mathrm{CB}}(d)$ of Fig. 3 is, in the range $3 \leqslant d \leqslant 3.84$, a monotonic curve consistent in shape and position with the curves extracted from the NPRG and the $\epsilon$-expansion. We have thus ruled out the S-shaped FD curve in Fig. 2, which has a turnaround point at $d \approx 3.2$ [14].[26]

Let us now come back to the question (3) about the existence of unitary stable CFTs with O($N$)×O(2) symmetry for $N = 2, 3$. Being fully agnostic, there still remains small loopholes which might allow their existence. First, on the basis of our results alone, we cannot rule out a turnaround of the $N_c(d)$ curve at some $d_\bullet < 3$, i.e. somewhat below the range explored here. Indeed, in the MZM fixed-dimensional scheme (see footnote 26), such a turnaround does happen for $d$ *slightly* below 3 (see the discussion in [14] below Fig. 3), although its precise location has not been determined. It seems implausible from the look of our curve in Fig. 3 that such a turnaround would happen just below $d = 3$. In the future it would be interesting to extend our study to smaller $d$ in order to definitively exclude such a turnaround.

---

[25]The PyCFTboot parameters `m_max` = 7, `n_max` = 9 in [25] correspond to a subset of $\Lambda = 2\,\texttt{n\_max}+\texttt{m\_max} = 25$ derivatives.

[26]The turnaround point $d \approx 3.2$ corresponds to the $\overline{\mathrm{MS}}$ fixed-dimension scheme, generally considered in the literature as more reliable than another scheme called MZM (massive zero-momentum). The fixed points found in the MZM scheme for $N = 2, 3$ lie outside of the region where the beta functions are Borel summable [11], whereas they are found around the boundary of this region in the $\overline{\mathrm{MS}}$ scheme [14].

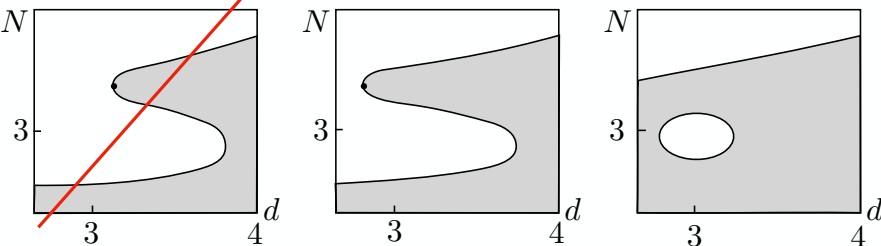

Figure 13: Three scenarios for the shape of the curve separating the region of the $(d, N)$ plane where a unitary $O(N) \times O(2)$ CFT exists (white) from the one where it does not (gray). The scenario on the left, where the boundary curve has a turnaround point (●) at a $d_\bullet > 3$, is excluded by our work. The two other scenarios, where the turnaround point is at a $d_\bullet < 3$, or where the allowed region is not connected, are still allowed.

Second, even if there is no turnaround, we cannot a priori exclude an allowed $(d, N)$ region which is disconnected from the region at large $N$. These two still allowed scenarios (as well as the excluded one) are illustrated in Fig. 13. We are however adamant that the CFTs in the allowed region for integer $d$ and $N$, being unitary, cannot be of focus type, but should have real correction-to-scaling exponent $\omega$.

## Acknowledgments

We thank Ning Su for discussions related to Remark 2.2, for sharing with us the `skydive` code before it was publicly released [50], as well as for the `simpleboot` framework software [75], and for his invaluable support in using these codes. SR thanks Yu Nakayama, Hugh Osborn, Marco Serone and Andreas Stergiou for discussions related to Appendix C, and Yin-Chen He for discussions related to Appendix A.

**Funding information** This work is supported by the Simons Foundation grants 733758 and 488659 (Simons Bootstrap Collaboration). A part of this work was completed during the Bootstrap 2023 program at the Instituto Principia (São Paulo). BvR acknowledges funding from the European Union (ERC "QFTinAdS", project number 101087025). Views and opinions expressed are however those of the authors only and do not necessarily reflect those of the European Union or the European Research Council Executive Agency. Neither the European Union nor the granting authority can be held responsible for them. The work of MR was also supported by the UK Research and Innovation (UKRI) under the UK government's Horizon Europe funding Guarantee (grant number EP/X042618/1), and the work of BS was also supported by a *Fonds de Recherche du Québec – Nature et technologies* B2X Doctoral scholarship. Some of the computations presented here were conducted in the Resnick High Performance Computing Center, a facility supported by Resnick Sustainability Institute at the California Institute of Technology.

## A  Frustrated magnets

The theoretical study of noncollinear magnets dates back to the 1970s ( [1, 76–80], reviewed e.g. in [81]). It is interesting to compare them to the usual ferromagnets described by the fer-

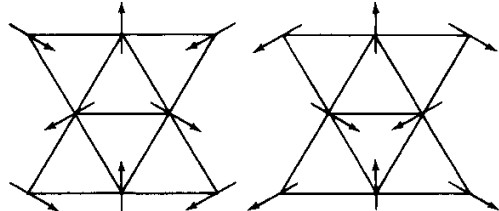

Figure 14: The ordered states of an STA (figure from [1]). The shown states have opposite chirality. There is a continuum of states obtained from the shown ones by a global rotation of all spins.

romagnetic O($N$) model ($N$-component spins at each vertex of a lattice, with nearest-neighbor ferromagnetic interactions). The usual ferromagnets have a disordered state at high $T$ and, in $d > 2$, an ordered state at low $T$, which breaks O($N$) spontaneously to O($N-1$) ("all spins point in the same direction"). The critical behavior is the standard O($N$) universality class, also known as the Wilson-Fisher O($N$) fixed point, or the O($N$) model CFT.

In contrast to the usual ferromagnets, noncollinear magnets have couplings between spins which involve some frustration. This leads to low-temperature states with a more complicated ordering. The symmetry breaking structure is different and the phase transition, if second order, is expected to be in a universality class distinct from O($N$).

Frustration is achieved by introducing some degree of antiferromagnetism, i.e. changing the sign of some nearest-neighbor couplings. On the cubic lattice with only nearest-neighbor couplings, every spin will be exactly anti-aligned with its neighbors at low-$T$, so there is no frustration and the phase transition is still in the O($N$) universality class. True frustration may appear on a lattice involving triangular faces.

One type of frustrated magnets which exhibits critical behavior distinct from O($N$) are the so-called *stacked triangular antiferromagnets* (STA). Spins are placed at the vertices of planar triangular 2d lattices stacked on top of one another in the third direction. Within each plane, nearest-neighbor interactions $J_\parallel$ are antiferromagnetic. The sign of interactions between the spins on nearby planes, denoted $J_\perp$, is not important. For continuous spins, possible ordered states at low-T will have the structure in each plane shown in Fig. 14, with spins at nearby sites rotated by 120°.

Frustration can also be obtained directly on the cubic lattice, by adding to the ferromagnetic O($N$) model a competing antiferromagnetic next-to-nearest neighbor interaction along a given direction. This construction models so-called *helical magnets*, and together with STAs, they have been referred to in the literature as *noncollinear magnets*.

Both systems near criticality enjoy the same continuum description. The procedure to obtain the Landau-Ginzburg-Wilson (LGW) Hamiltonian from a lattice model is straightforward and quite short (see Appendix A of [1]). This Hamiltonian is written in terms of the coarse-grained magnetization field $\vec{S}(x) \in \mathbb{R}^N$. The interaction, assumed to be a polynomial in $\vec{S}$, is expanded around every minimum of the quadratic part of the Hamiltonian density in momentum space. For the usual ferromagnets described by the O($N$) model, there is only one minimum in momentum space, located at $p = 0$. This leads to the well-known $\phi^4$-theory in terms of a single $N$-component field. In contrast, for noncollinear magnets, one finds two minima at two inequivalent Brillouin zone momenta $\pm Q$. The effective Hamiltonian is written in terms of a complex field $\vec{\varphi}(x) = e^{iQx}\vec{S}(x)$. O($N$) invariance and momentum conservation shows that the numbers $N$ and $N_*$ of $\vec{\varphi}$'s and $\vec{\varphi}^*$'s should satisfy the relation $N - N_* = 0$ mod 6.[27] This implies that the effective Hamiltonian has symmetry O($N$) × $\mathbb{Z}_6$. At the level of the quartic effective Hamiltonian, $\mathbb{Z}_6$ gets enhanced to a continuous O(2). One thus obtains

---

[27]To derive this relation, one observes that $3Q$ is in the reciprocal lattice.

the Hamiltonian (1) or, by writing $\varphi = a + ib$ where $a, b$ are real $N$ component fields,

$$\mathcal{H} = \frac{1}{2}\left((\partial_\mu a)^2 + (\partial_\mu b)^2\right) + r_0\left(a^2 + b^2\right) + u\left(a^2 + b^2\right)^2 + v\left((a \cdot b)^2 - a^2 b^2\right). \qquad \text{(A.1)}$$

One may wonder why such anisotropic systems as the STAs and helimagnets are described near criticality by the isotropic effective Hamiltonian (A.1)? Let us consider for example the case of STAs. After performing every step listed above in going from the microscopic to the effective Hamiltonian, one ends up with a kinetic term of the schematic form $\int d^d q\,(|J_\perp|q_\parallel^2 + |J_\perp|q_\perp^2)(a(q) \cdot a(-q) + b(q) \cdot b(-q))$, where $\parallel, \perp$ refers to directions inside of/perpendicular to the triangular lattice planes [82]. The anisotropy depending on the relative strength of the intra-plane and inter-plane couplings $J_\parallel$ and $J_\perp$ is then scaled away by rescaling momenta.

Let us describe the phase transitions expected in (A.1). As the mass $r_0$ is tuned, (A.1) undergoes a phase transition from the disordered state with $\langle a \rangle = \langle b \rangle = 0$ to a nontrivially ordered state whose structure crucially depends on the sign of the coupling $v$. Namely one finds $a \perp b$ for $v > 0$, while $a \parallel b$ for $v < 0$. The first possibility reproduces the noncollinear chiral behavior of Fig. 14, while the second describes so-called spin-density waves. In mean field theory the sign of the coupling $v$ is arbitrary, while in RG theory the sign of $v$ will be the one corresponding to the stable fixed point. Since noncollinear magnets have low-T ground states like in Fig. 14, we expect that the RG fixed point describing their phase transition (if second-order) should have $v > 0$. Indeed the stable fixed point $\mathcal{C}_+$ in Fig. 1 is located at $v > 0$.

# B Critical remarks about the focus fixed points

## B.1 Problems with unitarity

The main reason to doubt focus fixed points is that they contradict unitarity. Our model is unitary, i.e. reflection positive in the Euclidean signature. The critical points, if they exist, should be consistent with unitarity. This would imply that the scaling dimensions of scalar operators must be real. Instead, the focus fixed points have a complex correction-to-scaling exponent $\omega$. This means that there exists an operator $\mathcal{O}$ with a complex scaling dimension $\Delta = d + \omega$, whose two point function is given at large distances by

$$\langle \mathcal{O}(0)\mathcal{O}(x) \rangle \sim \frac{1}{r^{2\Delta}}. \qquad \text{(B.1)}$$

The complex conjugate operator $\mathcal{O}^*$ will have the complex conjugate correlator. Assuming conformal invariance (see Appendix C), only operators of equal scaling dimensions have non-vanishing 2pt functions, hence $\langle \mathcal{O}(0)\mathcal{O}^*(x) \rangle$ vanishes at large distances. Now consider a real operator $\Phi = \mathcal{O} + \mathcal{O}^*$. Its 2pt function is given by

$$\langle \Phi(0)\Phi(x) \rangle \sim \frac{1}{r^{2\Delta}} + \frac{1}{r^{2\Delta^*}} = \frac{2\cos(2b\log r)}{r^{2a}}, \qquad \text{(B.2)}$$

where $a = \text{Re}\Delta$, $b = \text{Im}\Delta$. If $b \neq 0$, there are distances at which $\langle \Phi(0)\Phi(x) \rangle < 0$. This contradicts reflection positivity.

## B.2 Merger curve and bifurcation theory

As discussed in the introduction, both the NPRG and the $\epsilon$-expansion predict a monotonic single-valued curve $N = N_c(d)$, while some fixed-dimension (FD) RG schemes predict a more complicated S-shaped curve (see Fig. 2).

As was emphasized by [69], the curve along which two fixed points (for us, $\mathcal{C}_+$ and $\mathcal{C}_-$) merge and annihilate (the merger curve) corresponds to a saddle-node bifurcation in the space of RG flows. This bifurcation is known to be a stable codimension-1 bifurcation for any number of parameters. In particular, for the 2-dimensional parameter space $(d, N)$, we expect the saddle-node bifurcation to occur, generically, along a smooth 1-dimensional line in the parameter space. Nothing can be predicted about the shape of this line on the basis of bifurcation theory alone. Our merger curve could as well be monotonic or S-shaped.

However, FD not only predicts the S-shape of the merger curve, but also focus fixed points. How can the appearance of focus fixed points be accommodated by bifurcation theory? A hint can be found in [83] which points out the existence on the merger curve of a special point $S$ where both eigenvalues of the stability matrix vanish: $\omega_1 = \omega_2 = 0$, whereas at generic points on this boundary only one eigenvalue vanishes. Using this hint, we identify the point $S$ as a Bogdanov-Takens (BT) bifurcation [84]. It is a codimension-2 bifurcation with the property that the stability matrix (for two dimensional flows) has the form:

$$\begin{pmatrix} 0 & A \\ 0 & 0 \end{pmatrix}, \qquad A \neq 0. \tag{B.3}$$

By a change of coordinates, the BT bifurcation flow can be brought to the normal form:

$$\dot{y}_1 = y_2, \tag{B.4}$$
$$\dot{y}_2 = \beta_1 + \beta_2 y_1 + y_1^2 + \sigma y_1 y_2,$$

where $\sigma = \pm 1$. In what follows we consider $\sigma = +1$.[28] Here $y_1, y_2$ should be thought of as coordinates in the $u, v$ space near the point $S$, while $\beta_1, \beta_2$ are coordinates in the parameter space $(d, N)$ near $(d_S, N_S)$. In the new coordinates, the point $S$ is realized for $\beta_1 = \beta_2 = 0$ and is located at $y_1 = y_2 = 0$.

The phase portraits of RG flows around the BT bifurcation are shown in Fig. 15.[29] To the right of the point $S$, at $\beta_1 > \beta_2^2/4$, we have region ①, in which there are no real fixed points. In terms of $N$ and $d$ this is the region delineated by the merger curve, on which $S$ is a distinguished point. Rotating counterclockwise, we cross the part of the merger curve labeled $T_+$ beyond which two fixed points appear, one stable and one unstable. So far this is just a saddle-node bifurcation, but if we go a bit further we cross a new curve $L_+$ on which a stable node-to-spiral transition happens (the two real eigenvalues $\omega_1, \omega_2$ collide and go into the complex plane).[30] In region ③ we have a saddle and a stable spiral. Rotating further still, we reach curve $P$, where a (global) saddle homoclinic bifurcation happens. Namely, on $P$ we have a homoclinic orbit which encircles the stable spiral fixed point and joins the saddle point to itself. On the other side of $P$, in region ④, the homoclinic orbit turns into an unstable cycle which limits the basin of attraction of the stable fixed point (which is still a spiral). This unstable cycle shrinks to a point on line $H$ (the Andronov-Hopf bifurcation), and in region ⑤

---

[28]$\sigma = -1$ is obtainable by flipping the sign of $y_2$ and $t$. Flipping the sign of $t$ would turn stable foci into unstable ones, which is not what we need.

[29]Many of these features were mentioned in [83], however the identification with the BT bifurcation is made here for the first time. Previously, BT bifurcations were found in perturbative RG flows of a scalar-fermion model in $d = 3 - \epsilon$ dimensions with $O(N) \times O(M)$ symmetry, $\phi^6$ interactions, and $\mathcal{N} = 1$ supersymmetry, for noninteger values of $N$ and $M$ [86].

[30]Near $S$, curve $L$ lies very close to curve $T$, its equation being

$$\beta_1 = 8\beta_2 + (16 - 2\beta_2)\sqrt{4 - \beta_2} - 32 = \beta_2^2/4 - \beta_2^4/1024 + O(\beta_2^5).$$

The distance between $L$ and $T$ was artificially increased in Fig. 15 for readability. It is significant that region ② (as well as region ⑥), where all exponents are real, is non-empty. Indeed, the direct annihilation of a spiral and a saddle is non-generic. The spiral has to first turn into a node, and only then annihilate. Region ② grows as one gets further away from $S$, see e.g. [87].

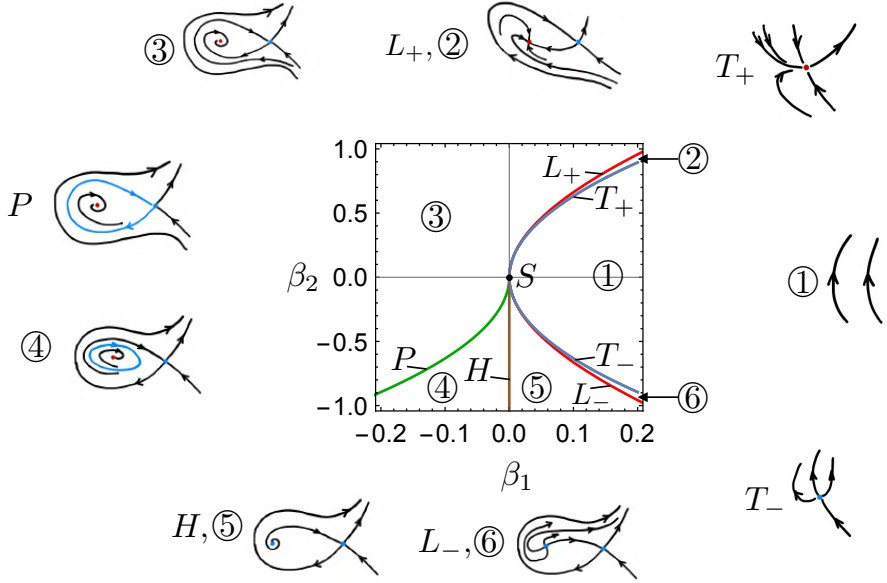

Figure 15: Phase portraits of RG flows in the vicinity of a BT bifurcation. See similar figures in [84] and [85], Fig. 137. Compared to [84], we added the lines $L_\pm$ which are the loci of the spiral to node transitions. (Note that these are technically not bifurcations but they are physically important.)

we have an unstable saddle and an unstable spiral. The unstable spiral turns into an unstable node along $L_-$. Finally, the unstable node and the saddle merge and annihilate on $T_-$.

## B.3 Further problems of focus fixed points

Using the previous subsection, we can point out the following further problems raised by focus fixed points. A basic principle of RG theory is the gradient flow property (see footnote 5), which says that the beta function equations should have the form

$$\beta^I = G^{IJ}\partial_J A, \tag{B.5}$$

where $A$ is a scalar function of the couplings and $G^{IJ}$ is a symmetric, positive-definite metric in the coupling space. This property is widely expected, although not yet proven. This property was originally found to hold up to three loops in a general multiscalar model in $d = 4 - \epsilon$ dimensions [15,16]. In $d = 4$ dimensions, a weaker form of the gradient flow property is proven to all orders of perturbation theory [88], where the metric $G^{IJ}$ may have an antisymmetric part starting from four loops (it is not known if the antisymmetric part actually occurs).[31] Later the stronger form of the gradient property, with a symmetric metric, was checked at four loops [89] and, very recently, at five and six loops both in $d = 4$ and $d = 4 - \epsilon$ dimensions [17], using the six-loop beta function computed in [18].

If the gradient flow property holds, the fixed points are identified with the critical points of the potential function $A$. The stability matrix at a fixed point is then given by:

$$\Gamma^I_K = \partial_K \beta^I = G^{IJ}\partial_J\partial_K A. \tag{B.6}$$

---

[31]It is also recognized that for the property to hold, the beta function should be computed in a particular "gauge", where it is known as the $B$-function [88]. The $B$-function fixed the beta-function ambiguity which exists in any multi-field theory, in which a renormalization group step can be accompanied by an infinitesimal rotation in the field space.

Although this stability matrix is not necessarily symmetric, it is a product of two symmetric matrices. The eigenvalue problem $\Gamma_K^I v^K = \lambda v^I$ for the stability matrix is equivalent to the generalized eigenvalue problem

$$(\partial_J \partial_K A) v^K = \lambda G_{JK} v^K \, . \tag{B.7}$$

This implies that all eigenvalues $\lambda$ must be real. This excludes focus fixed points.

Another consequence of the gradient flow property is that $A$ changes monotonically along the RG flow:[32]

$$\dot{A} = -\beta^I \partial_I A = -G^{IJ} \partial_I A \partial_J A < 0 \, . \tag{B.8}$$

This excludes cyclic RG flows. We have seen in the previous section that the RG flow diagram around a BT bifurcation contains RG cycles in region ④ - a further conflict.

Our final remark concerns the fact that focus fixed points appear when two correction-to-scaling critical exponents $\omega_1$, $\omega_2$ collide and then go to the complex plane. This happens along the curve $L$ in Fig. 15. Such level crossings between scaling dimensions of operators having the same symmetry are normally not allowed without finetuning. See [58, 90, 91] for recent discussions.

All of the above reasons lead us to believe that the focus type chiral fixed points found in [11–14] are artifacts of uncontrolled resummations. For previous exchanges on the controversy surrounding focus fixed points in the context of the O($N$) × O(2) model, see [4, 10, 92, 93].[33]

## C Conformal invariance of multiscalar models

In this appendix we recall why multiscalar fixed points are believed to be conformally invariant and not just scale invariant. The multiscalar fixed points in any $d$ are local, i.e. they have a local conserved stress tensor $T_{\mu\nu}$ of scaling dimension $d$. The condition for a local fixed point in $d > 2$ to be scale invariant without being conformal is that the trace of the stress tensor be a divergence of a local vector operator $V_\mu$ known as a virial current [95]

$$T^\mu_\mu = \partial_\mu V^\mu \, , \tag{C.1}$$

where $V^\mu \neq J^\mu + \partial_\nu L^{\mu\nu}$ with $J_\mu$ a conserved current ($\partial_\mu J^\mu = 0$) and $L_{\mu\nu} = L_{\nu\mu}$. Here and below, we do not keep track of the terms in $T^\mu_\mu$ which vanish by the equations of motion.

Since scaling dimensions of non-conserved vector operators are not protected, we do not generically expect an interacting fixed point to have a virial current candidate of precisely dimension $d-1$ so that it can appear in the r.h.s. of (C.1). Hence generically scale invariance in presence of interactions implies conformal invariance [96, 97].[34]

For multiscalar fixed points one can also give an argument not relying on the genericity assumption. The argument below is in perturbation theory in $d = 4-\epsilon$ dimensions (to all orders in $\epsilon$). Consider a multiscalar theory of $n$ real scalar fields with a general quartic interaction:

$$\frac{1}{4!} \lambda_{ijkl} \phi_i \phi_j \phi_k \phi_l =: \lambda^I \mathcal{O}_I \, . \tag{C.2}$$

Here we introduced the notation $\lambda^I$ for all couplings. The stress tensor trace along the RG flow has the form

$$T^\mu_\mu = \beta^I \mathcal{O}_I + \partial_\mu J^\mu_\nu + a_{ij} \partial^2 (\phi_i \phi_j), \tag{C.3}$$

---

[32]This would hold even if $G^{IJ} \to G^{IJ} + B^{[IJ]}$ in (B.5), as the antisymmetric part $B^{[IJ]}$ drops out.

[33]It has also been reported to us that these problematic fixed point also have unstable quartic potentials, which would be yet another reason to discard them [94]. We thank Marco Serone for discussions.

[34]Some interacting models with scale invariance without conformal invariance do exist [98–100]. They all have an additional noncompact "shift" symmetry which protects the virial current dimension [100].

where $\beta^I = d\lambda^I/d\log\mu$ is the beta function, and $J_\nu^\mu$ is a flavor current associated with the $O(n)$ Lie algebra element $\nu$, $J_\nu^\mu = \nu_{ij}\phi_i \overleftrightarrow{\partial_\mu} \phi_j$. The last term $a_{ij}\partial^2(\phi_i\phi_j)$ corresponds to "improvable terms" [95] and we will omit it in the discussion below since by itself it does not preclude the theory from being conformal.

At the fixed point we have $\beta_I = 0$. It looks like the theory may not be conformal if the second term $\partial_\mu J_\nu^\mu$ appears in (C.3) with a $\nu$ corresponding to a broken generator of $O(n)$. We do not know of a completely general argument to all orders in $\epsilon$ that this may not happen. For arguments at low orders in perturbation theory see [95,101].[35]

An all-order argument can however be given for fixed points with a sufficiently large symmetry group $G \subset O(n)$. Namely, suppose that the representation in which the broken currents transform does not contain a singlet. Then, since the stress tensor is a singlet, the broken currents cannot appear in the r.h.s. of (C.3), and scale invariance implies conformal invariance.

For the $O(N) \times O(2)$ model studied in this paper, the broken currents transform in the bifundamental representation, and the above argument applies.

# D  1/N expansion

We review here the $1/N$ expansion of model (1) [1,102–104], and compute the leading $1/N$ correction to the dimension of ST′. The Lagrangian in the form relevant to the large-$N$ expansion reads [104]

$$\mathcal{L} = \frac{1}{2}(\partial_\mu\phi)^2 + \frac{1}{2}\sigma\phi_{ai}\phi_{ai} + \frac{1}{2}T_{ij}\phi_{ai}\phi_{aj} - \frac{\sigma^2}{16u+4v} - \frac{1}{8v}\operatorname{Tr}T^2\,, \tag{D.1}$$

where we introduced two auxiliary fields $\sigma$ and $T_{ij}$. They are scalars belonging to the SS and ST irreps, and they morally replace $\phi^2$ and $\phi_{ai}\phi_{aj} - \frac{1}{2}\phi^2\delta_{ij}$. They have scaling dimensions

$$\Delta_\sigma, \Delta_{T_{ij}} = 2 + O(1/N)\,, \tag{D.2}$$

in any $d$, in contrast with the classical dimension $\Delta = d-2$ of the fields they replace. Scaling dimensions and OPE coefficients may be obtained as expansions in $1/N$, the coefficients of these expansions being computable by standard diagrammatic techniques.

At large $N$, all four fixed points on the left of Fig. 1 exist. They are distinguished by which of the auxiliary fields propagate: at $\mathcal{G}$, of course none does; at $\mathcal{H}$, only $\sigma$ does, leading to the well known $1/N$ expansion of the $O(N)$ model; at $\mathcal{C}_-$, only $T_{ij}$ propagates; and finally, both auxiliary fields propagate at $\mathcal{C}_+$. The difference in field content between $\mathcal{C}_+$ and $\mathcal{H}$ leads to an appreciable difference between the large-$N$ spectrum of both theories, the most obvious differences being in the scalar ST sector. At $\mathcal{C}_+$, $\mathcal{O}_{\text{ST}} = T_{ij}$, followed by two subleading fields $\mathcal{O}_{\text{ST}'}$ and $\mathcal{O}_{\text{ST}''}$ with $\Delta = 4 + O(1/N)$ resulting from the mixing of $(T^2)_{ij} - \frac{1}{2}\operatorname{Tr}T^2\delta_{ij}$ and $\sigma T_{ij}$. In contrast, $T_{ij}$ is absent at $\mathcal{H}$, so that $\mathcal{O}_{\text{ST}} = \phi_{ai}\phi_{aj} - \frac{1}{2}\phi^2\delta_{ij}$, followed by $\mathcal{O}_{\text{ST}'} = \sigma\mathcal{O}_{\text{ST}}$ with $\Delta = d + O(1/N)$.

Although imposing a gap of the order of $d-2$ in the ST channel might have been enough to differentiate between both theories, we have found more constraining power by assuming the existence of $\mathcal{O}_{\text{ST}}$, and imposing a gap above it. To motivate the value of this gap, it helps to know the $1/N$ correction to the dimension of $\mathcal{O}_{\text{ST}'}$ in $\mathcal{H}$ and $\mathcal{C}_+$. In $\mathcal{H}$, this operator descends from the subleading symmetric traceless 2-tensor operator of $O(N)$, which has been computed long ago in [105]:

$$\Delta_{\text{ST}'} = \Delta_{\text{ST}'}^{(1)} + O(1/N^2)\,, \qquad \Delta_{\text{ST}'}^{(1)} = d - \frac{2\left(d^2 - 3d + 4\right)\Gamma(d-1)}{\Gamma\left(1 - \frac{d}{2}\right)\Gamma\left(\frac{d}{2} + 1\right)\Gamma\left(\frac{d}{2}\right)^2}\frac{1}{N}\,. \qquad (\mathcal{H}) \tag{D.3}$$

---

[35]S.R. thanks Yu Nakayama, Hugh Osborn and Andy Stergiou for discussions.

In $\mathcal{C}_+$, this computation requires resolving the mixing problem between $\mathcal{O}_{\text{ST}'}$ and $\mathcal{O}_{\text{ST}''}$. Ref. [106] gives a nice and concise description of the procedure to follow to compute anomalous dimensions of composite operators at large-$N$. The example provided in this paper lists and explicitly computes all diagrams contributing to the anomalous dimension of $\sigma^2$ in the $O(N)$ model. Since $\mathcal{C}_+$ differs from $\mathcal{H}$ by the addition of a new $T_{ij}$ propagator as well as a new $T\phi\phi$ vertex [102]:

$$T_{ij}\phi_{ak}\phi_{bl} \rightarrow \delta_{ab}K_{ijkl},$$
$$K_{ijkl} = \frac{1}{2}\left(\delta_{ik}\delta_{jl} + \delta_{il}\delta_{jk} - \delta_{ij}\delta_{kl}\right), \tag{D.4}$$

it should be clear that we can recycle some of the results of [106]. Indeed, the diagrams contributing to the renormalization of $(T^2)_{ij} - \text{trace}$ and $\sigma T_{ij}$ will have the same geometry as those contributing to $\sigma^2$. Here is an example of such a diagram arising in the computation of the anomalous dimension of $\sigma^2$:

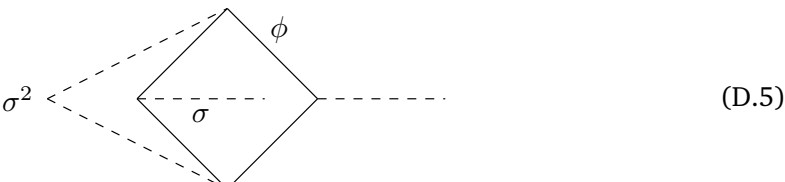

$$\tag{D.5}$$

One can think of this diagram as computing the renormalization of $\sigma^2$ by considering the three point function $\langle\sigma^2\sigma\sigma\rangle$. Diagram (D.5) has the following two nonzero cousins in $\mathcal{C}_+$ :

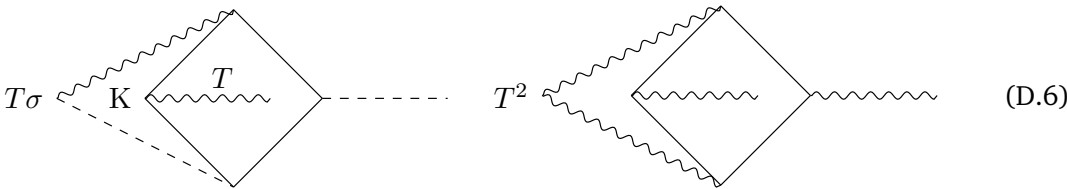

$$\tag{D.6}$$

These are determinable from Diagram (D.5) up to $O(2)$ group structure factors, coming from $T_{ij}$ propagators and $T\phi\phi$ vertices, and differences in symmetry factors. There is luckily no need to compute any loop integrals. After the dust settles, the leading order anomalous dimensions of this mixing problem are given by[36]

$$\gamma_1 = \frac{4(12 - 22d + 13d^2 - 2d^3)\sin\left(\frac{\pi d}{2}\right)\Gamma(d-2)}{\pi d\,\Gamma\left(\frac{d}{2}-1\right)\Gamma\left(\frac{d}{2}\right)}\frac{1}{N},$$
$$\gamma_2 = \frac{2^{d-3}(24 - 28d + 5d^2 + d^3)\sin\left(\frac{\pi d}{2}\right)\Gamma\left(\frac{d-1}{2}\right)}{\pi^{3/2}\Gamma\left(\frac{d}{2}+1\right)}\frac{1}{N}. \tag{D.7}$$

The scaling dimensions of $\mathcal{O}_{\text{ST}'}$ in $\mathcal{C}_+$ is then given by

$$\Delta_{\text{ST}'} = \Delta_{\text{ST}'}^{(1)} + O(1/N^2), \qquad \Delta_{\text{ST}'}^{(1)} = 4 + \min(\gamma_1, \gamma_2) + O(1/N^2). \tag{D.8}$$

# E  $\epsilon$-expansion results

## E.1  $N_c(d)$

In the $\epsilon$-expansion, the location $(u^*, v^*)$ of fixed points and critical exponents in $d - 4 - \epsilon$ are obtained as asymptotic series expansions in $\epsilon$. To locate the critical value of $N$ where $\mathcal{C}_+$

---

[36]We checked this result by comparing it at $O(\epsilon, 1/N)$ to the anomalous dimensions of the corresponding operators in the $\epsilon$-expansion. These do not seem to have been previously computed, but they are easily obtained using the OPE method described e.g. in [54].

merges and annihilates with $\mathcal{C}_-$, one can ask that the location of both fixed points coincide:

$$u_-^*(\epsilon, N_c(4-\epsilon)) = u_+^*(\epsilon, N_c(4-\epsilon)),\tag{E.1}$$
$$v_-^*(\epsilon, N_c(4-\epsilon)) = v_+^*(\epsilon, N_c(4-\epsilon)).\tag{E.2}$$

Equivalently, one can look for the value of $N$ at which the critical exponent $\omega$, related to the scaling dimension of the subleading SS scalar, is zero:

$$\omega_\pm(\epsilon, N_c(4-\epsilon)) = 0.\tag{E.3}$$

In both cases, one obtains $N_c(4-\epsilon)$ as a series expansion in $\epsilon$. We have ([6], Table I, first line)[37]

$$N_c(4-\epsilon) = 21.798 - 23.431\epsilon + 7.0882\epsilon^2 - 0.0321\epsilon^3 + 4.2650\epsilon^4 - 8.4436\epsilon^5 + O(\epsilon^6).\tag{E.4}$$

This series is divergent, and to obtain the best possible estimate of $N_c(d)$ one would need to resum it. There are many ways to do so, see [6] for investigations on how to do so optimally. Here we used the method of [6], Eqs. (25),(26) which goes back to [107], Eq. (7). Rather than working with the original series, we first input the expected $d=2$ behavior:

$$N_c(4-\epsilon) = 2 + (2-\epsilon)a(\epsilon),\tag{E.5}$$

and then expand $\frac{1}{a(\epsilon)}$. The coefficients of the series for $\frac{1}{a(\epsilon)}$ are better behaved than those of the original series, and we sum it directly, using the contribution of the last term as the error estimate. The so obtained $N_c(d)$ will be denoted $N_c^\epsilon(d)$. This method is simple and gives results in good agreement with other methods, based on Padé and Borel resummation as in [6]. This procedure was also used to compare the $\epsilon$-expansion to the results of the NPRG in [108].

## E.2 Scaling dimensions

In several parts of the paper we referred to the values of the critical exponents evaluated in the $\epsilon$-expansion. We first explain the general methodology used in the literature [6, 66, 102].

The $\epsilon$-expansion is easiest performed for $N > N_c(4) = 4(3+\sqrt{6})$. In this range of $N$ the fixed point couplings, and therefore the critical exponents, can be expanded as (asymptotic) power series in integer powers of $\epsilon$ with real, $N$-dependent, coefficients. However in several places in this paper we need the critical exponents for $N < N_c(4)$. In this range of $N$ we cannot simply use the series computed for $N > N_c(4)$, since those series have some coefficients proportional to $\sqrt{N - N_c(4)}$, and become complex for $N < N_c(4)$.

At the formal level, what is happening is that the assumption of expanding the couplings in integer powers of $\epsilon$ breaks down for $N = N_c(4) - O(\epsilon)$. One can see for example that for $N = N_c(4-\epsilon)$ given by (E.4) the fixed point couplings have an expansion in half-integer powers. On general grounds, we expect that for $N = N_c(4-\epsilon) + \delta N$ with $\delta N > 0$, the critical exponents should have dependence on $\delta N$ involving positive half-integer powers. We can determine this expansion in $\delta N$ by substituting $N = N_c(4-\epsilon) + \delta N$ into the series derived for $N > N_c(4)$, and reexpand in $\epsilon$. This was proposed below Eq.(3.29) of [102], referred there as a "trick", although it's not a trick but a mathematically legitimate way to match two asymptotic expansions with overlapping ranges of validity. As a consistency check, one observes that all terms involving negative half-integer powers of $\delta N$, which could appear a priori when expanding $\sqrt{\delta N + O(\epsilon)}$ in $\epsilon$, cancel order by order in the expansion in $\epsilon$.

---

[37]The first two terms are given exactly by $12 + 4\sqrt{6} - (12 + 14\sqrt{2/3})\epsilon$. This is useful for checking the cancellation of $1/\sqrt{\delta N}$ singularities in the reexpansion of critical exponents discussed in Appendix E.2.

One thus obtains critical exponents as a series in $\epsilon$ (with integer powers) with coefficients real functions of $\delta N$ with at most positive square-root singularities at $\delta N = 0$. Substituting, for a given $N$, $\delta N = N - N_c^\epsilon(d)$ where $N_c^\epsilon(d)$ is as in Appendix E.1, we get critical exponents as a series in $\epsilon$ with numerical coefficients. For small $\epsilon$, these series can be used by directly summing the first few terms. For larger $\epsilon$, such as in $d = 3$, these series need to be Borel or Padé resummed to be useful.

We now give precise references to sources for each figure.

In Fig. 5, to position $\mathcal{H}, \mathcal{C}_\pm$ we used expansion of critical exponents listed in the ancillary file of [25]. This corresponds to order $\epsilon^3$ for $\Delta_\phi$ and to order $\epsilon^2$ for $\Delta_{SS}$ and $\Delta_{ST}$. In their conventions:

- $\Delta_\phi$ is `deltaPhiChiralEps` ($\mathcal{C}_+$), `deltaPhiAntiEps` ($\mathcal{C}_-$), `deltaPhiONEps` ($\mathcal{H}$)

- $S \equiv SS$ and $W \equiv ST$

- $\Delta_{SS}$, $\Delta_{ST}$ is `deltaScalarsChiralEps[[i]]` ($\mathcal{C}_+$), `deltaScalarsAntiEps[[i]]` ($\mathcal{C}_-$), `deltaScalarsONEps[[i]]` ($\mathcal{H}$), where $i = 1, 2$ for $\Delta_{SS}, \Delta_{ST}$.

As described above, we substitute $N = N_c(4 - \epsilon) + \delta N$, reexpand in $\epsilon$, and subsitute $\delta N = N - N_c^\epsilon(d)$. Since we are at small $\epsilon = 0.2$, the resulting series is simply summed.

In Fig. 7, we obtained, as described in the previous paragraph, $\Delta_\phi, \Delta_{SS}$ and $\Delta_{ST}$ at the $\mathcal{C}_+$ fixed point as a series in $\epsilon$, to order $\epsilon^2$ for $\Delta_{SS}$ and $\Delta_{ST}$ and to order $\epsilon^3$ for $\Delta_\phi$, with coefficients depending on $\delta N$. We plugged $\epsilon = 0.2$ and plotted the resulting function, too bulky to report here, as a function of $\delta N = N - N_c^\epsilon(3.8)$.

In Fig. 6, since we are dealing with $\epsilon = 1$, the $\epsilon$-expansion series from [25] are too short to be useful, and anyway they would have to be resummed. Fortunately for $\mathcal{C}_+$ this has been done elsewhere in the literature, so we borrow the available results. Thus, for $\mathcal{C}_+$, we extract the scaling dimensions of $\phi$ and SS from the $n = 8$ "Final" values of $\eta = 0.042(2)$ and $\nu = 0.745(11)$ in Table 9 of [6]. These critical exponents are related to the scaling dimensions as $\Delta_\phi = \frac{d-2}{2} + \frac{\eta}{2}$ and $\Delta_{SS} = d - \frac{1}{\nu}$. $\Delta_{ST}$ for $\mathcal{C}_+$ was obtained from the critical exponent $y_4 = 1.13(8)$ in the $\overline{\text{MS}}$ scheme of Table III of [66]. The two are related as $\Delta_{ST} = 3 - y_4$. We have not found in the literature equally precise resummed estimates for $\mathcal{C}_-$, or for $\mathcal{H}$ (at $N = 8$ that we need for Fig. 6). For them, we used the large-$N$ results at the highest known order. For $\mathcal{C}_-$, this is $1/N$ order, and the results are tabulated in the ancillary file of [25]. For $\mathcal{H}$, $\Delta_\phi$ is known to order $1/N^3$ [109], and the other two to order $1/N^2$ [110, 111]. The three are nicely tabulated in the ancillary file of [112].

In Fig. 6, since we are dealing with $\epsilon = 1$, the $\epsilon$-expansion series has to be resummed to be useful. Fortunately for $\mathcal{C}_+$ this has been done elsewhere in the literature, so we borrow the available results. Thus, for $\mathcal{C}_+$, we extract the scaling dimensions of $\phi$ and SS from the $n = 8$ "Final" values of $\eta = 0.042(2)$ and $\nu = 0.745(11)$ in Table 9 of [6]. These critical exponents are related to the scaling dimensions as $\Delta_\phi = \frac{d-2}{2} + \frac{\eta}{2}$ and $\Delta_{SS} = d - \frac{1}{\nu}$. $\Delta_{ST}$ for $\mathcal{C}_+$ was obtained from the critical exponent $y_4 = 1.13(8)$ in the $\overline{\text{MS}}$ scheme of Table III of [66]. The two are related as $\Delta_{ST} = 3 - y_4$. We have not found in the literature equally precise resummed estimates for $\mathcal{C}_-$, or for $\mathcal{H}$ (at $N = 8$ that we need for Fig. 6). For these cases, we instead used the large-$N$ results at the highest known order. For $\mathcal{C}_-$, this is $1/N$ order, and the results are tabulated in the ancillary file of [25]. For $\mathcal{H}$, $\Delta_\phi$ is known to order $1/N^3$ [109], and the other two to order $1/N^2$ [110, 111]. The three are nicely tabulated in the ancillary file of [112].

# F Numerical bootstrap details

## F.1 The navigator method

The central problem in the numerical conformal bootstrap is classifying parameter space into allowed and disallowed regions. With the navigator function technology [49] this is done by splitting the problem into two steps. First step: *look for an allowed point by minimizing the navigator $\mathcal{N}(x)$ from a starting point $x_0$*. In Appendix F.4 we will show how to construct a navigator function for the class of numerical conformal bootstrap problems that we consider in this paper. We recall that this function should be negative for allowed values of $x$, positive for disallowed values, and zero at the boundaries between these regions. It should also be differentiable.

Once an allowed point $x_*$ is found, the second step is: *look for the boundaries of the allowed region around $x_*$ which is the zero set of the navigator*. Both steps can be phrased as non-convex optimization problems, and algorithms for solving them were given in [49]. A particular second-step problem concerns finding the maximal extent of the allowed region in a fixed direction, which can be phrased as

$$\text{maximize } n^T x \text{ over all } x \text{ such that } \mathcal{N}(x) \leqslant 0. \tag{F.1}$$

Here $n$ is a vector describing the direction in parameter space that we want to maximize, while still remaining in the same isolated allowed region to which $x_*$ belongs.

As in the main text, we generally suppress the dependence of the navigator function on any parameters that are held fixed in the optimization process (like $d$ and $N$ in some cases).

## F.2 Details for Figs. 5 and 6

The above strategy is precisely how the islands in Figs. 5 and 6 were "boxed in." First we located an allowed point in each isolated allowed region, by minimizing $\mathcal{N}(x)$ starting from many initial starting points. Subsequently, within each islands we determined the minimal and maximal allowed values for all three parameters $\Delta_\phi$, $\Delta_{SS}$, $\Delta_{ST}$ that the navigator function depends on.

We stress again that the optimizations problems encountered in both steps are non-convex. In the first step, the navigator $\mathcal{N}(x)$ has multiple minima. All navigator minima which we observed in our searches, starting from many initial points, fell into one of the following three categories:

1. A negative minumum in one of the isolated allowed regions corresponding to an allowed physical CFT (like $\mathcal{H}$ or $\mathcal{C}_+$).

2. A negative minimum near the location of a CFT that is (barely) excluded by the bootstrap assumptions, like $\mathcal{C}_-$ in Section 3.1.

3. As the parameters $N$ and $\text{gapST}_0$ vary, some of the isolated allowed regions may disappear, but the corresponding minima may still remain, although they are now positive.

In this sense, all seen navigator minima were related to either a CFT satisfying the bootstrap assumptions or a CFT that "nearly" obeys the bootstrap assumptions.

In the second step, when finding the boundaries of the islands, non-convexity again plays a role. By extremizing from multiple different points and in additional directions $\vec{n}$ we found that at certain derivative orders $\Lambda$ the isolated allowed regions are not convex. Specifically for some $\Lambda$ and for some $\text{gapST}_0$, we detected a narrow "bridge" between the $\mathcal{C}_+$ and $\mathcal{H}$ islands. However, we ascertained that for the $\Lambda$ and $\text{gapST}_0$ used in Fig. 5 and Fig. 6 no such bridge

Table 2: Parameters for `skydive`.

| | |
|---|---|
| $\Lambda$ | 31 |
| $\varkappa$ | 30 |
| order | 120 |
| spins | $\{0,\dots,40\} \cup \{46, 47, 51, 52, 55, 56, 59, 60\}$ |
| precision | 768 |
| dualityGapThreshold | $10^{-20}$ |
| primalErrorThreshold | $10^{-15}$ |
| dualErrorThreshold | $10^{-15}$ |
| initialMatrixScalePrimal | $10^{20}$ |
| initialMatrixScaleDual | $10^{20}$ |
| feasibleCenteringParameter | 0.3 |
| infeasibleCenteringParameter | 0.3 |
| stepLengthReduction | 0.7 |
| maxComplementarity | $10^{100}$ |
| dualityGapUpperLimit | 0.2 |
| externalParamInfinitestimal | $10^{-40}$ |
| findBoundaryObjThreshold | $10^{-20}$ |
| betaScanMin | 0.3 |
| betaScanMax | 1.01 |
| betaScanStep | 0.1 |
| stepMinThreshold | 0.1 |
| stepMaxThreshold | 0.6 |
| primalDualObjWeight | 0.2 |
| navigatorWithLogDetX | False |
| navigatorAutomaticShift | False |
| gradientWithLogDetX | True |
| betaClimbing | 1.5 |
| optimalbeta | False |
| stickToGCP | False |

exists. To check this we extremized from different starting points and in additional directions $\vec{n}$, including from starting points in each isolated allowed region in the direction of the other allowed regions.

### F.3 The `skydive` algorithm

For initial explorations of plots like in Figs. 5 and 6, up to $\Lambda = 27$, we used Algorithm 1 (for the first step) and Algorithm 2 (for the second step) from [49]. For the final computations at $\Lambda = 31$ we instead used the `skydive` algorithm [50]. Both of these methods use a BFGS-like quasi-Newton method. The algorithms from [49] required computing the exact navigator function $\mathcal{N}(x)$ at each point $x$ along the minimization path. The `skydive` algorithm operates instead with an approximate navigator function $\mathcal{N}_\mu(x)$ where $\mu$ is an internal SDPB parameter which can be thought of as describing the accuracy of the approximation. We have $\lim_{\mu \to 0} \mathcal{N}_\mu(x) = \mathcal{N}(x)$. In `skydive`, $\mu$ is lowered gradually as the minimization progresses. As a result the total minimization cost in `skydive` is comparable to a single evaluation of the navigator function. This algorithm offered a significant speed-up, allowing us to use $\Lambda = 31$ for most of our results, and even use $\Lambda = 43$ in some checks.

In `skydive`, the navigator function and its derivative are sometimes estimated poorly in the early steps when $\mu$ is large, which can degrade the performance. To mitigate this, it is

essential to choose an appropriate value for the `dualityGapUpperLimit`. This value sets an upper limit on $\mu$ for each step, which ensures that the approximation of the navigator function is good enough to guide the search towards the minimum of the true navigator function $\mathcal{N}(x)$. Setting this value too small leads to a slow search, while setting it too large often results in the search steering off towards the allowed "peninsula" at large external dimensions, despite being in the region of attraction of another minimum of the true navigator. In the first step problem, we found the value of 0.05 to be effective for searches starting relatively far from the allowed point, while 0.005 is appropriate for searches starting close to the navigator minimum (the same holds for searches of the second step).[38]

The complete workflow was as follows. We used `scalar_blocks` [113] to generate conformal blocks; `simpleboot` [75] to setup the semi-definite problems and to manage the computations on the cluster; either SDPB [35,37] to solve the SDPs (i.e. to compute the navigator function) and BFGS-like Algorithms 1 and 2 from [49] (implemented in `simpleboot`) to move in the parameter space, or the new `skydive` to perform the last two steps at once [50]. These codes depend on a number of parameters that must be set by the user, the most crucial being the derivative order $\Lambda$ that dictates the dimension of the space of functionals SDPB will search in. For the most important computations at $\Lambda = 31$ we used `skydive`, with the parameters given in Table 2.

## F.4 Constructing the navigator function

In order to start applying the navigator method in Section 3.1, we have to first make a choice of the particular navigator function we are going to use. Ref. [49] introduced two different navigator constructions: the GFF navigator and the $\Sigma$ navigator. For the entirety of the paper, we chose to work with the GFF navigator, which has an advantage of a natural normalization and an upper bound: $\mathcal{N}(x) \leqslant 1$. Say we are considering a given set of crossing equations

$$\vec{E}(x) = 0, \tag{F.2}$$

depending on a set of parameters $x$, usually scaling dimensions or ratios of OPE coefficients. We add a term $\lambda \vec{M}_{\mathrm{add}}$ to this crossing equation such that the augmented crossing equation

$$\vec{E}(x) + \lambda \vec{M}_{\mathrm{add}} = 0, \tag{F.3}$$

always has a solution for some positive $\lambda$. The navigator function is defined [49] as the minimal value of $\lambda$ for which a solution exists. Different terms $\vec{M}_{\mathrm{add}}$ lead to different navigator functions. The GFF navigator function is obtained by adding the contribution of every operator below the assumed gaps from a solution where each external field is an independent generalized free field (GFF). This idea is completely general and was already used to study the Ising model [49] and the O($N$) model [60].

Here we use it for O($N$) × O(2) symmetric CFTs. We have chosen the additional terms to correspond to the solution to crossing where $\phi$ is an O($2N$) GFF, and $s$ is another, independent GFF. The O($2N$) invariant GFF solution for $\phi$ also has the symmetry of its O($N$) × O(2) subgroup, and thus solves the O($N$) × O(2) crossing equation. To explicitly write $\vec{M}_{\mathrm{add}}$ in this case, we need to know how operators in the O($2N$) invariant solution decompose under O($N$)×O(2). The nontrivial decomposition rules (including the weights in the conformal block decomposition) are:

$$T \to \frac{1}{2}TT + \frac{1}{2}AA + \frac{1}{2}TS + \frac{1}{N}ST, \tag{F.4a}$$

$$A \to \frac{1}{2}AT + \frac{1}{2}TA + \frac{1}{2}AS + \frac{1}{N}SA. \tag{F.4b}$$

---

[38]This choice of course strongly depends on the normalization of the navigator functions that is used. Here we are assuming a navigator function that has been normalized to be bounded by 1.

The terms in the r.h.s. are the irreps XY of O($N$)×O(2) appearing in the decomposition of irreps $T, A$ of O($2N$). The coefficients multiplying them are the relative weights needed to pass from the O($2N$) invariant conformal block decomposition of $\langle\phi\phi\phi\phi\rangle$ to the O($N$) × O(2) invariant one. These are obtained as follows. Consider the O($2N$) generalized free field $\phi_I \equiv \phi_{ai}$. Its 4pt function decomposes into contributions of the different O($2N$) irreps $\mathcal{R}$ present in the $\phi_I \times \phi_J$ OPE, weighted by tensor structures:

$$\langle\phi_I\phi_J\phi_K\phi_L\rangle \equiv \langle\phi_{ai}\phi_{bj}\phi_{ck}\phi_{dl}\rangle = \mathrm{T}^{\mathrm{S}}_{IJKL}G_{\mathrm{S}} + \mathrm{T}^{\mathrm{T}}_{IJKL}G_{\mathrm{T}} + \mathrm{T}^{\mathrm{A}}_{IJKL}G_{\mathrm{A}}; \tag{F.5}$$

$$\mathrm{T}^{\mathrm{S}}_{IJKL} = \delta_{IJ}\delta_{KL} \equiv \delta_{ij}\delta_{kl}\delta_{ab}\delta_{cd},$$
$$\mathrm{T}^{\mathrm{A}}_{IJKL} = \delta_{IK}\delta_{JL} - \delta_{IL}\delta_{JK} \equiv \delta_{ik}\delta_{jl}\delta_{ac}\delta_{bd} - \delta_{il}\delta_{jk}\delta_{ad}\delta_{bc}, \tag{F.6}$$
$$\mathrm{T}^{\mathrm{T}}_{IJKL} = \delta_{IK}\delta_{JL} + \delta_{IL}\delta_{JK} - \frac{2}{2N}\delta_{IJ}\delta_{KL} \equiv \delta_{ik}\delta_{jl}\delta_{ac}\delta_{bd} + \delta_{il}\delta_{jk}\delta_{ad}\delta_{bc} + \frac{2}{2N}\delta_{ij}\delta_{kl}\delta_{ab}\delta_{cd}.$$

We obtain the branching rules (F.4) by decomposing these tensor structures in terms of products $\mathrm{T}^{\mathrm{X}}_{ijkl}\mathrm{T}^{\mathrm{Y}}_{abcd}$, where XY is an irrep of O($N$) × O(2) appearing in the decomposition of $\mathcal{R}$.

## F.5 Minimum-following for Fig. 3

As explained in Section 3.2, finding the minimal allowed value of $N$ for Fig. 3 can be seen as a second-step optimization problem for the navigator function depending on $N$ as the extra fourth parameter. We used the `skydive` algorithm at $\Lambda = 31$ to solve this problem.

The initial points for the `skydive` algorithm were obtained using an auxiliary run at $\Lambda = 19$, using the following different strategy inspired by [60], which we call "minimum-following." In this strategy we first minimize the navigator function with respect to the parameters $x = (\Delta_\phi, \Delta_{\mathrm{SS}}, \Delta_{\mathrm{ST}})$. We then track the location of this minimum as we vary $N$ and $d$.

In this section we write $\mathcal{N}(x, N, d)$ to emphasize that our algorithm also moves us around in $N$ and $d$.

For example, suppose we start at an allowed value of $p$ where $N \gg N_c(d)$, so at the minimum $x_{\min}(N, d)$ we find that the navigator function is negative:

$$\mathcal{N}(x_{\min}(N, d), N, d) < 0. \tag{F.7}$$

If we now decrease $N$ we can follow this minimum until we reach $N_c$, where $\mathcal{N} = 0$. Then, once we have found this $N_c$ at some value of $d$, we may follow this extremum $N_c$ as we change $d$ from say $d$ near 4 to $d = 3$.

When we have minimized the navigator function over its internal parameters $x$, with $N$ close to $N_c(d)$, we can predict the shape of $N_c(d)$ by solving the following equation

$$\mathcal{N}(x_{\min} + \delta x_{\min}, N + \delta N, d + \delta d) = 0 \tag{F.8}$$

to first order in $\delta N$ and $\delta d$. Note that the linear term in $\delta x_{\min}$ vanishes because we are at the navigator minimum. For example, if we are at a fixed $d$ (so $\delta d = 0$), we can step towards $N_c(d)$ by taking the step

$$\delta N = -\frac{\mathcal{N}(x_{\min}, N, d)}{\frac{\partial \mathcal{N}}{\partial N}(x_{\min}, N, d)}, \tag{F.9}$$

which is the Newton step following from the first-order expansion of (F.8). The partial derivative in the denominator can be computed via finite differences or with the SDP gradient formula of Section 4 of [49]. Then, once we have determined $N_c$ at this $d$, we can take a step $\delta d$. To

remain on the critical curve $N_c(d)$ we should take a corresponding step $\delta N = N_c'(d) \times \delta d$, where from (F.8):

$$N_c'(d) = -\frac{\frac{\partial \mathcal{N}}{\partial d}(x_{\min}, N, d)}{\frac{\partial \mathcal{N}}{\partial N}(x_{\min}, N, d)}. \tag{F.10}$$

This minimum-following method has advantages and disadvantages which make it complementary to the direct method based on extremizing the 4-variable navigator island in the $N$ direction. By construction, it sticks close to the navigator minima, thus exploring less parameter space. This is both a positive, since the searches are less likely to be attracted by another unimportant navigator minimum, such as the peninsula, but also a downside, since it might miss important minima not connected smoothly enough to previous minima. The direct method proved less robust, as it sometimes makes too large a step and escapes from the region of interest (the $\mathcal{C}_+$ island) to disconnected regions like the $\mathcal{H}$ island or the peninsula. However, when successful, the direct method is more computationally efficient. Also, the direct method can be parallelized, running separate computation for several $d$'s, while the minimum-following method cannot be easily parallelized as it explores the shape of the curve $N_c(d)$ locally in $d$.

In the end, both methods proved useful. We first used minimum-following to find the entire curve $N_c(d)$ for $\Lambda = 19$ and some particular set of gap assumptions.[39] To move to higher $\Lambda$ as in Fig. 3, we used the allowed points of zero navigator obtained from minimum-following to initialize parallel `skydive` computations of $N_c(d)$ for the 9 values $d = 3.0, 3.1, \ldots, 3.8$.

### F.6 Gap sensitivity

We finish this appendix by reporting some tests concerning the impact of gap assumptions (7b)-(7d) on $N_c(d)$. Since we have chosen them very conservatively, $N_c(d)$ should not depend much on them. This is indeed what we observe. More precisely, we see that varying these gaps has a moderate effect on the shapes of the island, but a much smaller effect on the value of $N_c(d)$. E.g. we see that $N_c^{\text{CB}}$ in $d = 3.8$ varies by 0.02 when varying the gaps above the stress tensor and the conserved currents between 0.7 and 1.2 (these gaps are set to 1 in (7b)-(7d)).

A similar remark concerns the `gapVV`$_0$ assumption (9). We observed (again in $d = 3.8$) hardly any effect on $N_c^{\text{CB}}(d)$ when we varied `gapVV`$_0$ between 1.7 to 2.5. On the other hand the negative navigator minimum disappeared when we set this gap to a much larger value `gapVV`$_0 = 3.2$. This behavior is consistent with the estimates $\Delta_{\text{VV}'} \approx 2.7 - 2.9$ in $d = 3.8$ from (8).

Finally, the dependence of the bounds in $d = 3$ on `gapST`$_0$ was discussed around Fig. 9 in the main text.

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
