# Peer review of "Bootstrapping frustrated magnets: the fate of the chiral ${\rm O}(N)\times {\rm O}(2)$ universality class"

_SciPost Physics, doi:SciPost Phys. 18, 060 (2025)_

## Round 2 · Referee Report · Anonymous (Referee 1) · 2024-7-28

Report

This paper considers the merger and annihilation of CFTs with O(N)xO(2) symmetry in d=3, which describe certain statistical systems. Using the conformal bootstrap, they find an island of an allowed region that matches perturbative results at large N and near d=4, and they compute how this island disappears as a function of d and N. In particular, they find that it disappears for d=3 for N>3, which excludes the most physically interesting cases of N=2,3. The island disappears likely due to violation of unitarity as the CFT goes into the complex plane, this violation of unitarity is much more significant than the violation of unitarity from being in fractional N and d, which as they discuss is not significant enough to affect bootstrap results as shown in previous work. Near the point where the island disappears, they also find that CFT data shows a square root behavior as expected from merger and annihilation.

The paper is excellently written and all calculations seem correct, so I suggest no changes. Looking ahead, it would be nice to also bootstrap the model with two relevant singlets (this paper only focussed on the model with one relevant singlet), so that they can explicitly see the merger and annihilation of the two fixed points.

Recommendation

Publish (surpasses expectations and criteria for this Journal; among top 10%)

---

## Round 2 · Referee Report · Anonymous (Referee 2) · 2024-9-23

Report

The authors study phase transitions in the frustrated magnets, which are described by a matrix model with symmetry O(N)XO(2). They use the conformal bootstrap method and find that no unitary theory can be found in $d=3$ for N smaller than some value that they evaluate at 3.78.

The results obtained in this article are interesting and deserve publication in scipost. There is however few points I would like to raise.

1) In sect. 3.3.2 the authors study the fate of the fixed points C+ and C-, which they relate to minima of their navigator function, as N is decreased. They find that C+ "turns into a saddle". As far as I understand, the navigator function is continuous and there is a finite island where it is negative. Consequently, if C+ becomes a saddle point, local minima must appear at the same time (in agreement with Morse theory). What happens to these minima?

2) If the minimum C+ disapears around N=7, I would expect that, for $3.76<N<7, the island found by the authors corresponds to an unstable fixed point in the renormalization-group sense. The phase transition would be generically of first order except at some tricritical point and Nc(3) would be close to 7, not 3.76. Unfortunately, the authors are unable to compute the scaling dimension of the operator SS' and cannot really test this. An alternative would be that it is not always true that a local minimum can be associated with a fixed point. I would encourage the authors to comment on this.

3) In Fig 8, the curves representing the scaling dimensions of various operators clearly show a feature around 4.2 (a breaking of the slope or maybe a square root). What is at the origin of this behavior?

4) Finally, I think that the references 45 and 46 should appear earlier in the introduction.

Recommendation

Publish (meets expectations and criteria for this Journal)

---

## Round 3 · Referee Report · Anonymous (Referee 1) · 2024-10-17

Report

The paper is now ready for publication.

Recommendation

Publish (easily meets expectations and criteria for this Journal; among top 50%)

---

## Round 3 · Referee Report · Anonymous (Referee 2) · 2024-11-21

Report

The revised version is fine for publication. I still don't understand the author's answer concerning the saddle point but it is a matter of details and should not delay the publication.

I think the toy model that the authors propose is not valid because it does show a minimum turning into a saddle but instead a minimum annihilating with a maximum. Indeed, a saddle point is an extremum with negative Hessian (that would be a maximum in the 1d toy model). A toy model for a minimum turning into a saddle point is the mexican hat where the quadratic term changing sign. When the minimum becomes a saddle point (a local maximum in the 1d case), extra minima occur.

Recommendation

Publish (easily meets expectations and criteria for this Journal; among top 50%)

---

## Round 3 · Author Response

With this resubmission we have taken into account the feedback given by the referees and addressed their concerns. A full list of changes is given below.

In addition we would like to provide the following replies to some of the remarks in the reports:

  • Report 2 mentions "if C+ becomes a saddle point, local minima must appear at the same time (in agreement with Morse theory)." We are not aware of mathematical results which could prevent one of two minima in the negative region turning into a saddle in a one-parameter family of functions. Perhaps the following 1d example could help: The function f(x)=-1-x^2+x^4 has two symmetric minima, but deforming it by a linear term +h x the minimum at positive x turns into a saddle at h~1, while the minimum at negative x remains there. This is roughly what we believe happens as we decrease N, with C+ minimum turning into a saddle and disappearing, C- minimum remaining, and no new minima appearing.

  • Report 2 mentions "If the minimum C+ disappears around N=7, I would expect that, for $3.76<N<7, the island found by the authors corresponds to an unstable fixed point in the renormalization-group sense." While the C+ minimum disappears, we do not interpret this as the fixed point disappearing here. Note that there is no exact correspondence between navigator minima and fixed points. What is more important is the conformal data values that such a stable CFT would take cannot be excluded. I.e. they are in the allowed region. It is quite possible that at higher derivative order the C+ minima would persist (and would be separated from other allowed regions such as the C- island and the peninsula). However, proving this expectation would require further analysis beyond what is reported in the paper. We mostly think our conservative language already reflects the fact that the identification is only tentative, but we clarify this further with the following edit:

"We observe several different features compared to Fig. 7. First, at N = 8 the navigator function now has two well separated minima, close to the respective expected positions of C± within the island. We track both minima which we therefore tentatively call the C+ navigator minimum (diamonds) and the C− minimum (dots)."

and the addition of the footnote:

"While it is very interesting to note that all local minima of the navigator function discovered up to now seem to have their origin in physical theories it is important to keep in mind that the absence of the "C+" minimum below N<7 does not imply this theory does not exist. What is more important is that the conformal data values that such a stable CFT would have are not excluded by our current analysis for N>3.78, but are rigorously excluded for N<3.78."

-Report 2 asks whether we understand the origins of the features at small N in figure 8. We restructured the paragraph and added a footnote that clarifies that there is not sufficient evidence to interpret these features:

"Instead we see in \cref{fig:trackingD3} that the $\mathcal{C}_+$ minimum disappears too soon, followed by a rather large region between $N_c^\text{CB}(3) = 3.78$ and $N\approx 6.5$ without good agreement with the $\epsilon$-expansion. All of this could however be numerical artifacts, and in future work it would be interesting to see how the \cref{fig:trackingD3} evolves when increasing $\Lambda$ and whether the features at small $N$ remain or the bounds converge closer to the ideal scenario.\footnote{At this point it is for example unclear whether there is physical significance to the change of slopes in the conformal data that is observed around $N=4.2$. This would be interesting to investigate further.}"

  • Report 2 suggests to mention two references earlier which we have done.

---

## Round 3 · List of Changes

• p3: Added a footnote mentioning related previous numerical conformal bootstrap studies earlier when the numerical conformal bootstrap is first introduced.
  • p17: Added the word tentatively and restructured the sentence: "We observe several different features compared to Fig. 7. First, at N = 8 the navigator function now has two well separated minima, close to the respective expected positions of C± within the island. We track both minima which we therefore tentatively call the C+ navigator minimum (diamonds) and the C− minimum (dots)."
  • p17: Added the footnote: "While it is very interesting to note that all local minima of the navigator function discovered up to now seem to have their origin in physical theories it is important to keep in mind that the absence of the "C+" minimum below N<7 does not imply this theory does not exist. What is more important is that the conformal data values that such a stable CFT would have are not excluded by our current analysis for N>3.78, but are rigorously excluded for N<3.78."
  • p 18: Restructured a paragraph: "Instead we see in \cref{fig:trackingD3} that the $\mathcal{C}_+$ minimum disappears too soon, followed by a rather large region between $N_c^\text{CB}(3) = 3.78$ and $N\approx 6.5$ without good agreement with the $\epsilon$-expansion. All of this could however be numerical artifacts, and in future work it would be interesting to see how the \cref{fig:trackingD3} evolves when increasing $\Lambda$ and whether the features at small $N$ remain or the bounds converge closer to the ideal scenario." and added the footnote: "At this point it is for example unclear whether there is physical significance to the change of slopes in the conformal data that is observed around $N=4.2$. This would be interesting to investigate further." -p 25: Added an acknowledgement to the Resnick High Performance Computing Center, a facility supported by Resnick Sustainability Institute at the California Institute of Technology were some of the computations were performed.

---

## Round 4 · Author Response

We thank the referee for bringing this issue again to our attention, as indeed our wording was still not entirely precise. We hope the current version fully clarifies what is happening.

---

## Round 4 · List of Changes

-Replaced "At this point the local minimum turns into a saddle, which we noticed by explicitly computing the Hessian, and the navigator function itself is still negative so we are within the island"
with:
"At this point the local minimum collides with a saddle, leading to the disappearance of both. Precisely at this transition, one Hessian eigenvalue is zero, which we checked by explicitly computing the Hessian. However, the navigator function itself is still negative so we are within the island" (edited) .

-Updated grant acknowledgement according to new guidelines.

---

## Editorial Decision

published